# GenDA: Generative Data Assimilation on Complex Urban Areas via Classifier-Free Diffusion Guidance

**Francisco Giral** [1]  **Álvaro Manzano Sevillano** [1]  **Ignacio Gomez Perez** [1]  **Ricardo Vinuesa** [2]  **Soledad Le Clainche** [1]

## Abstract

Urban wind flow reconstruction is essential for assessing air quality, heat dispersion, and pedestrian comfort, yet remains challenging when only sparse sensor data are available. We propose GenDA, a generative data assimilation framework that reconstructs high-resolution wind fields on unstructured meshes from limited observations. The model employs a multiscale graph-based diffusion architecture trained on computational fluid dynamics (CFD) simulations and interprets classifier-free guidance as a learned posterior reconstruction mechanism: the unconditional branch learns a geometry-aware flow prior, while the sensor-conditioned branch injects observational constraints during sampling. This formulation enables obstacle-aware reconstruction and generalization to held-out mesh geometries, wind directions, and sensor configurations within the studied urban-flow setting, without retraining. We consider both sparse fixed sensors and trajectory-based observations using the same reconstruction procedure. When evaluated against supervised graph neural network (GNN) baselines and classical reduced-order data assimilation methods, GenDA reduces the relative root-mean-square error (RRMSE) by 25-57% and increases the structural similarity index (SSIM) by 23-33% across the tested meshes. Experiments are conducted on Reynolds-averaged Navier-Stokes (RANS) simulations of a real urban neighborhood in Bristol, United Kingdom, at a characteristic Reynolds number of $\mathrm{Re} \approx 2 \times 10^7$, featuring complex building geometry and irregular terrain. The proposed framework provides a scalable path toward gener-

ative, geometry-aware data assimilation for environmental monitoring in complex domains.

## 1. Introduction

Urban wind flow plays a pivotal role in determining the microclimate, pollutant dispersion, and pedestrian comfort in densely populated areas. With cities becoming increasingly instrumented through fixed and mobile environmental sensors, there is an urgent need to reconstruct complete, high-resolution wind fields from sparse and partial measurements. Applications span from air quality monitoring and heat stress mapping to wind safety in street canyons and building-integrated energy systems (Torres et al., 2021). Urban turbulent flows exhibit strong nonlinearity, multi-scale vortex interactions, and sensitivity to localized building geometry (Torres et al., 2021; Nithya et al., 2024), making simple interpolation or physics-only modeling approaches unreliable when sensor density is low or spatial coverage is heterogeneous. Furthermore, high-fidelity computational fluid dynamics (CFD) simulations remain computationally costly for large urban domains, which limits their suitability for real-time environmental monitoring or iterative scenario assessments.

Classical data assimilation (DA) strategies such as the Ensemble Kalman Filter and reduced-order model (ROM)-based approaches integrate physical models with sensor observations to reconstruct latent flow fields (Jeanney et al., 2025). While effective in principle, their computational tractability typically relies on low-dimensional representations, either linear subspaces (e.g., proper orthogonal decomposition) or learned nonlinear latent manifolds (Solera-Rico et al., 2024; Eivazi et al., 2022), where the flow state is represented in a reduced space. This compression enhances computational efficiency but removes explicit geometric and obstacle-induced structural information from the representation (Xiang et al., 2021), making the learned basis specific to a particular domain configuration. When the underlying geometry changes, the reduced basis and associated observation operators must be adapted or recomputed (Hetherington & Le Clainche, 2025). Moreover, DA performed directly on CFD solvers can require repeated forward evaluations or adjoint-based sensitivity calcula-

---

[1]Applied Mathematics Department, ETSIAE-School of Aeronautics, Universidad Politécnica de Madrid, Spain [2]Department of Aerospace Engineering, University of Michigan, Ann Arbor, MI 48109, USA. Correspondence to: Francisco Giral <fa.giral@alumnos.upm.es>.

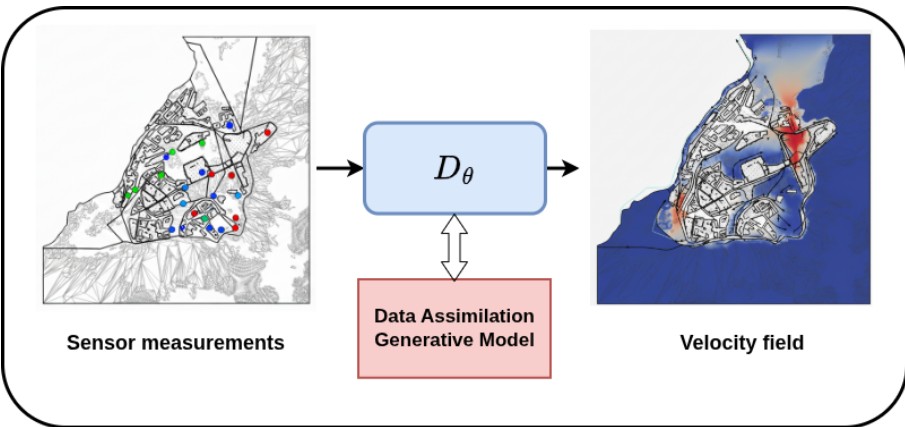

*Figure 1.* Sparse sensor measurements provide partial information on the flow, and the generative model $D_\theta$ reconstructs a physically plausible velocity field consistent with the observations and the urban geometry.

tions, significantly increasing computational cost in high-resolution urban settings (Moldovan et al., 2021; Plogmann et al., 2024). Reduced-order DA techniques such as low-cost singular value decomposition (SVD)-based assimilation can partially mitigate this (Hetherington & Le Clainche, 2025), but still rely on careful data preprocessing and sensor placement design.

In parallel, recent work has explored reconstruction of flow fields from sparse or incomplete sensor observations using machine learning, including supervised surrogates, physics-informed neural networks, and graph-based simulators (Gao et al., 2024; Liu et al., 2023; Guastoni et al., 2021). However, such models typically produce deterministic predictions and do not naturally incorporate new observations at inference time. For example, Gao et al. (Gao et al., 2024) employ a physics-informed graph-assisted autoencoder to reconstruct urban wind fields from sparse sensors, but the learned mapping remains fixed after training, making it difficult to adapt to new sensor layouts or previously unseen geometries. In urban contexts, Diff-SPORT (Vishwasrao et al., 2025) introduced diffusion-based reconstruction and sensor placement strategies in structured meshes.

Generative diffusion models have recently emerged as powerful tools for modeling the distribution of turbulent flows (Guastoni & Vinuesa, 2025). These models synthesize physically plausible velocity fields by reversing a noise corruption process, and naturally enable ensemble-based uncertainty quantification (Gutha et al., 2025). Applications include flow super-resolution (Oommen et al., 2025; Amorós-Trepat et al., 2026; Gao et al., 2025), obstacle-conditioned wake generation (Hu et al., 2025), and latent-space turbulent synthesis (Du et al., 2024). Building upon this direction, recent work on geometry-conditioned flow generation (Giral et al., 2025) demonstrated that diffusion models can learn a geometry-aware prior distribution over urban wind fields.

In the present work, we extend this idea to a full data assimilation setting by introducing a generative posterior reconstruction framework. Our model, GenDA, operates on unstructured meshes and incorporates sensor observations through classifier-free guidance. Building on the concept of geometry-conditioned priors (Giral et al., 2025), the unconditional branch of our diffusion model learns the underlying distribution of urban wind fields shaped by building-induced flow structures, while the sensor-conditioned branch learns to inject local observational constraints. During sampling, classifier-free guidance blends these two signals, acting effectively as a learned data assimilation operator: the unconditional branch provides a geometry-aware physical prior, and the conditional branch performs a posterior correction that steers samples toward consistency with available measurements. This interpretation enables obstacle-aware flow reconstruction that adapts to held-out mesh geometries, wind directions, and sensor layouts within the studied urban-flow distribution without retraining.

The contributions of our work are summarized as follows.

- We formulate sparse-observation urban wind reconstruction on unstructured meshes as a conditional generative data-assimilation problem and address it using a graph-based diffusion model.

- We interpret classifier-free guidance as a posterior reconstruction mechanism that balances a learned, geometry-aware prior with sparse sensor observations.

- We introduce a multiscale graph architecture that enables efficient local and global propagation of observational information across complex urban geometries.

- We empirically validate the proposed framework on realistic urban flow simulations, demonstrating improved accuracy and robustness over supervised GNN base-

lines, classical reduced-order data assimilation methods, and a stochastic diffusion baseline across held-out mesh slices, observation densities, sensor layouts, and held-out sensor interpolation tests.

The resulting framework supports flexible sensing configurations, including sparse fixed networks, building-mounted arrays, and mobile trajectories, and produces uncertainty-aware ensembles rather than deterministic estimates. While this work focuses on steady two-dimensional slices from an urban CFD dataset, the generative assimilation formulation suggests natural extensions to volumetric (3D) domains and to real-time air quality monitoring pipelines in cities, where mobile and heterogeneous sensors provide intermittent and spatially uneven observations. Empirical validation of these extensions is left for future work.

Our implementation is available in the code link.

This paper is organized as follows. Section 2 presents the proposed framework, model architecture, and training strategy. Section 3 provides qualitative and quantitative evaluations under different sensing scenarios. Finally, Section 4 summarizes the findings and discusses directions for future research.

## 2. Method

We propose a generative data assimilation framework that reconstructs full urban flow fields from sparse sensor measurements using a graph-based diffusion model with classifier-free guidance (CFG). Reconstruction is posed as an inverse problem: given an unstructured mesh $\mathcal{G}$, the wind direction $\Phi$, and a sparse set of sensor readings $\mathcal{S}$, we sample from the posterior of physically plausible velocity fields that match the observations. The wind direction $\Phi$ is treated as a known global descriptor of the physical operating condition, analogous to an inflow boundary condition; in practice, it can be obtained from meteorological measurements or upstream sensors. Figure 1 summarizes the high-level objective of the method, where partial observations across the urban area are used by the generative model to reconstruct the full flow field. Details on the diffusion formulation, training setup, and hyperparameters are provided in Appendix B.1 and Appendix B.3.

### 2.1. Problem Setup and Sensor Strategies

We evaluate our framework on an industrial-scale dataset derived from high-fidelity RANS simulations of a complex urban neighbourhood in Bristol, UK. The domain features a realistic building layout with a maximum height of 98 m. The flow is highly turbulent, characterized by a Reynolds number of Re $\approx 2.08 \times 10^7$ (based on a reference velocity of 3.18 m/s at 10 m height).

To adapt the volumetric flow to a graph-based learning setting, we extract horizontal slices at six different altitudes ($z \in \{15, 20, 28, 35, 40, 45\}$ m). Figure 2 shows the full 3D structure and an example of how a slice is taken. Although all slices originate from the same neighbourhood, each altitude yields a distinct graph with different fluid-solid interactions and boundary configurations, leading to systematically different flow patterns. We utilize four altitude slices for training and hold out two for testing, ensuring the model must generalize to previously unseen geometric configurations. Because the dataset consists of steady RANS fields, there is no temporal axis for a time-based split. Holding out altitude slices therefore tests transfer to unseen graph structures and slice-dependent obstacle configurations, rather than interpolation over repeated samples on the same mesh. Further details on the dataset parameters are provided in Appendix A.

**Multiscale Graph Representation** To enable efficient message passing across multiple spatial scales, each mesh is paired with a coarsened (reduced) version obtained offline through optimal decimation, using the algorithm of (Schroeder et al., 1992). In our dataset, the original 2D meshes contain on the order of $\sim 3 \times 10^5$ nodes and $\sim 1.7 \times 10^6$ edges, whereas the reduced meshes contain $\sim 5.5 \times 10^4$ nodes and $\sim 2.8 \times 10^5$ edges, corresponding to an approximate 5-6× reduction in nodes and edges. This yields a two-level multiscale graph structure with four edge sets:

- *o2o (original-to-original):* message passing between neighboring nodes on the original fine mesh, capturing local geometry and flow interactions;

- *o2r (original-to-reduced):* projection of encoded node features from the fine mesh to the coarsened mesh;

- *r2r (reduced-to-reduced):* message passing on the reduced mesh, enabling efficient long-range information propagation across the domain;

- *r2o (reduced-to-original):* projection back to the fine mesh, reconstructing high-resolution predictions.

To preserve obstacle boundaries, wall and domain-boundary vertices are retained with higher priority during decimation, ensuring that the reduced mesh maintains the building footprints and outer boundary shape. This hierarchy accelerates global communication while keeping predictions defined on the original-resolution mesh. In practice, this coarsened level acts as a latent communication graph that enables deeper message passing without exceeding memory limits: most processor layers operate on the reduced mesh, while only a small number of layers operate on the full-resolution graph. In our experiments, introducing the reduced mesh

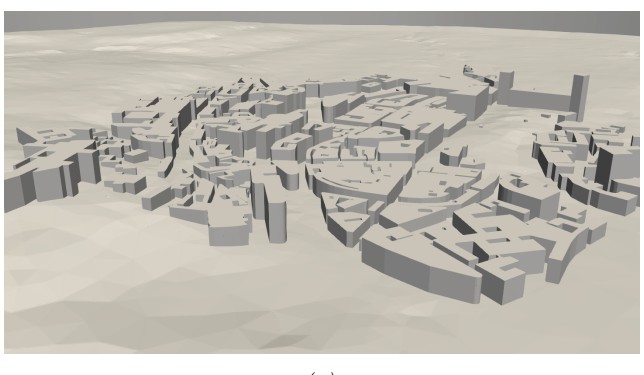 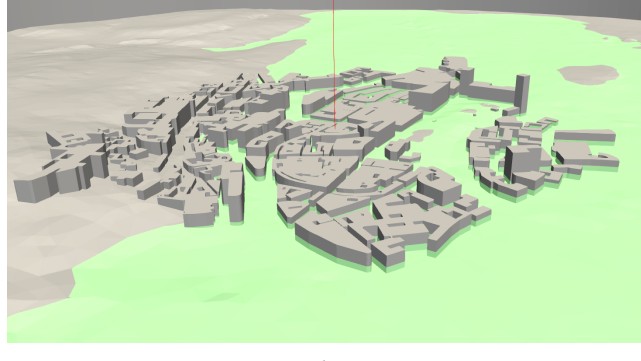

*(a)*        *(b)*

*Figure 2.* (a) 3D geometry of the modeled Bristol's neighbourhood, including buildings and underlying terrain. (b) Example of a horizontal slice normal to the vertical ($z$) axis, resulting in a 2D unstructured mesh used for graph-based diffusion. More information can be found at https://modelair.eu.

does not harm reconstruction accuracy, as reflected by the comparable performance of the multiscale supervised baseline relative to the single-scale MeshGraphNet baseline.

Figure 3 illustrates the multiscale mesh hierarchy used by the proposed framework. The original mesh represents the high-resolution discretization on which sensor data and predictions are defined. A reduced mesh is obtained offline through optimal decimation, preserving the global geometry while reducing node density. Message-passing operations are performed within and between these two meshes: *o2o* connections exchange information between neighboring nodes on the fine mesh, capturing local flow structures; *o2r* projections transfer encoded features from the fine to the reduced mesh; *r2r* edges propagate messages on the reduced mesh, enabling efficient long-range communication; and *r2o* projections return aggregated information back to the fine mesh for high-resolution reconstruction. This process allows the network to combine detailed geometric awareness with efficient global context exchange.

**Sensor Sampling Strategies** To reflect different practical sensing deployments, we consider multiple ways of selecting observed nodes. In all cases, observations are encoded by a binary mask $m_i$ and value $y_i$ at node $i$:

- *Random sampling:* nodes are drawn uniformly across the mesh, representing sparse or opportunistic sensor placement.

- *Dense sensor clouds:* a set of seed nodes is selected and expanded to their $k$-hop neighborhoods to emulate targeted, high-density measurement campaigns in specific regions.

- *Mobile trajectories:* nodes lying along paths (e.g., street networks) are selected to emulate mobile or vehicle-mounted sensors; for steady flows we aggre-

gate along these paths to form structured sparse observations.

These configurations allow us to assess the robustness to observation density, clustering patterns, and spatial coverage, critical for real deployments where sensor availability is limited and heterogeneous.

### 2.2. Graph-based generative data assimilation via tempered posterior sampling

We formulate urban flow reconstruction from sparse sensors as a *generative data assimilation* problem on unstructured meshes. Given a mesh $\mathcal{G}$, inflow direction $\Phi$, and sparse observations $\mathbf{y}$, the goal is to generate velocity fields that are simultaneously (i) physically plausible given the urban geometry and (ii) consistent with the available measurements. GenDA achieves this by combining a multiscale graph-based diffusion model with classifier-free guidance (CFG), yielding an efficient approximation to posterior sampling in high-dimensional, geometry-dependent domains.

**Multiscale graph architecture for data assimilation.** A central challenge in urban data assimilation is propagating sparse, localized observations across complex geometries while preserving sharp, obstacle-induced flow features. The proposed multiscale graph architecture addresses this by explicitly coupling local and global information pathways. Message passing on the original (fine) mesh captures local flow-geometry interactions near buildings, walls, and narrow passages, where gradients and recirculation are strongest. In parallel, message passing on a reduced (coarsened) mesh enables efficient long-range communication across the domain, allowing sparse observations to influence dynamically connected but spatially distant regions. Bidirectional projections between the fine and reduced meshes ensure that global context informs local corrections and vice versa. This hierarchy allows GenDA to assimilate sparse measurements with-

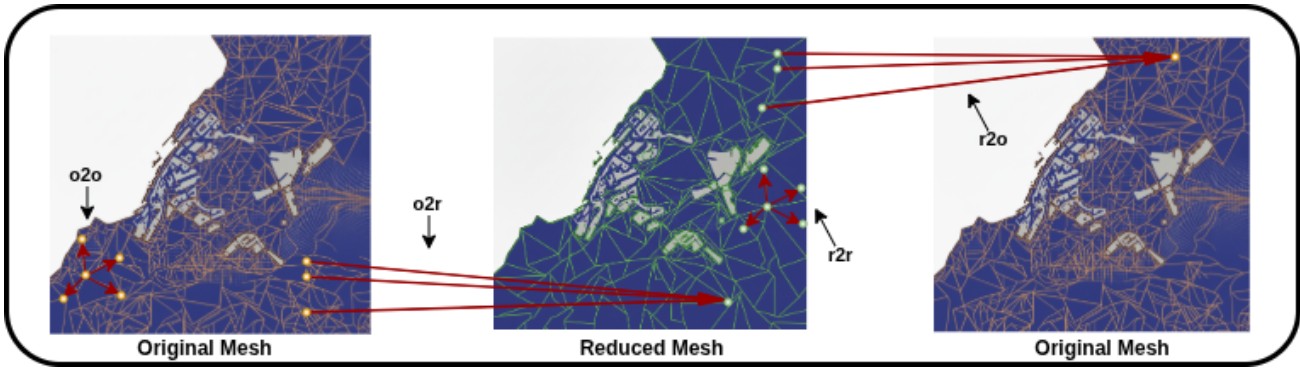

*Figure 3.* Multiscale mesh hierarchy used in the proposed framework. The original node-dense mesh (left) provides the high-resolution representation where sensor observations and predictions are defined. The reduced mesh (center) is obtained through optimal decimation and supports efficient long-range message passing.

out excessive smoothing or iterative global solvers, which are common limitations of classical reduced-order or variational methods. Further architectural details, including graph message-passing operations and node/edge inputs, are provided in Appendix B.2.

**Posterior formulation.** Let $\mathbf{u}^*$ denote the unknown velocity field over $\mathcal{G}$, and let observations be given by

$$\mathbf{y} = \mathcal{H}(\mathbf{u}^*) + \boldsymbol{\eta}, \tag{1}$$

where $\mathcal{H}$ selects observed nodes and $\boldsymbol{\eta}$ denotes measurement noise. In the experiments, observations are placed at mesh nodes to enable controlled benchmarking across sensor layouts. More general off-mesh measurements can be incorporated by defining $\mathcal{H}$ as an interpolation or projection operator from the continuous sensor locations to the mesh representation. Data assimilation seeks to characterize the posterior

$$p(\mathbf{u} \mid \mathbf{y}, \mathcal{G}) \propto p(\mathbf{u} \mid \mathcal{G}) \, p(\mathbf{y} \mid \mathbf{u}), \tag{2}$$

where $p(\mathbf{u} \mid \mathcal{G})$ represents a geometry-conditioned prior over physically plausible flows, and $p(\mathbf{y} \mid \mathbf{u})$ enforces agreement with sensor measurements. In GenDA, the prior $p(\mathbf{u} \mid \mathcal{G})$ is not prescribed analytically or represented by a fixed reduced basis. Instead, it is represented by the unconditional branch of the diffusion model directly on the graph, allowing the model to encode nonlinear, geometry-dependent flow structures such as wakes and recirculation regions.

Score-based diffusion models provide a mechanism for sampling from such distributions by following the gradient of the log-density. The score of the posterior in Eq. (2) decomposes additively into prior and observation terms,

$$\begin{aligned} \nabla_{\mathbf{u}} \log p(\mathbf{u} \mid \mathbf{y}, \mathcal{G}) &= \nabla_{\mathbf{u}} \log p(\mathbf{u} \mid \mathcal{G}) \\ &+ \nabla_{\mathbf{u}} \log p(\mathbf{y} \mid \mathbf{u}) + \text{const.}, \end{aligned} \tag{3}$$

which motivates the use of classifier-free guidance as an approximate posterior update mechanism.

**Classifier-free guidance as tempered posterior sampling.** Through sensor-dropout training, the diffusion model learns two related denoisers: an *unconditional* denoiser, which estimates the score of the geometry-conditioned prior $p(\mathbf{u} \mid \mathcal{G})$, and a *sensor-conditioned* denoiser, which incorporates measurement information. At inference time, CFG combines these two estimates as

$$\tilde{D}_\theta(\mathbf{u}, \sigma; \mathbf{y}) = D_\theta(\mathbf{u}, \sigma) + \gamma[D_\theta(\mathbf{u}, \sigma; \mathbf{y}) - D_\theta(\mathbf{u}, \sigma)], \tag{4}$$

where $\gamma$ controls the strength of guidance.

Figure 4 illustrates this process. At each reverse-diffusion step, the unconditional branch encodes a geometry-aware prior over urban flows, while the sensor-conditioned branch injects information from the sparse observations. Their difference acts as an observation-driven correction to the prior score, and the guidance weight $\gamma$ modulates how strongly this correction influences the sampling trajectory. Lower values of $\gamma$ favor samples drawn primarily from the learned physical prior, while larger values increasingly enforce measurement consistency.

This guided update can be interpreted as sampling from a *tempered posterior*, i.e., a posterior in which the observation likelihood is raised to a power $\gamma$,

$$p_\gamma(\mathbf{u} \mid \mathbf{y}, \mathcal{G}) \propto p(\mathbf{u} \mid \mathcal{G}) \, p(\mathbf{y} \mid \mathbf{u})^\gamma, \tag{5}$$

where $\gamma$ plays the role of an inverse temperature on the likelihood. In this view, $\gamma{=}0$ recovers unconditional prior sampling, $\gamma{=}1$ corresponds to standard conditioning, and $\gamma{>}1$ increases enforcement of observational constraints. This role of $\gamma$ is directly analogous to classical data assimilation, where the relative weighting between background and observation terms (e.g., via their assumed error covariances) controls how strongly measurements correct the prior state. The effect of $\gamma$ on reconstruction quality is empirically analyzed in Appendix C.

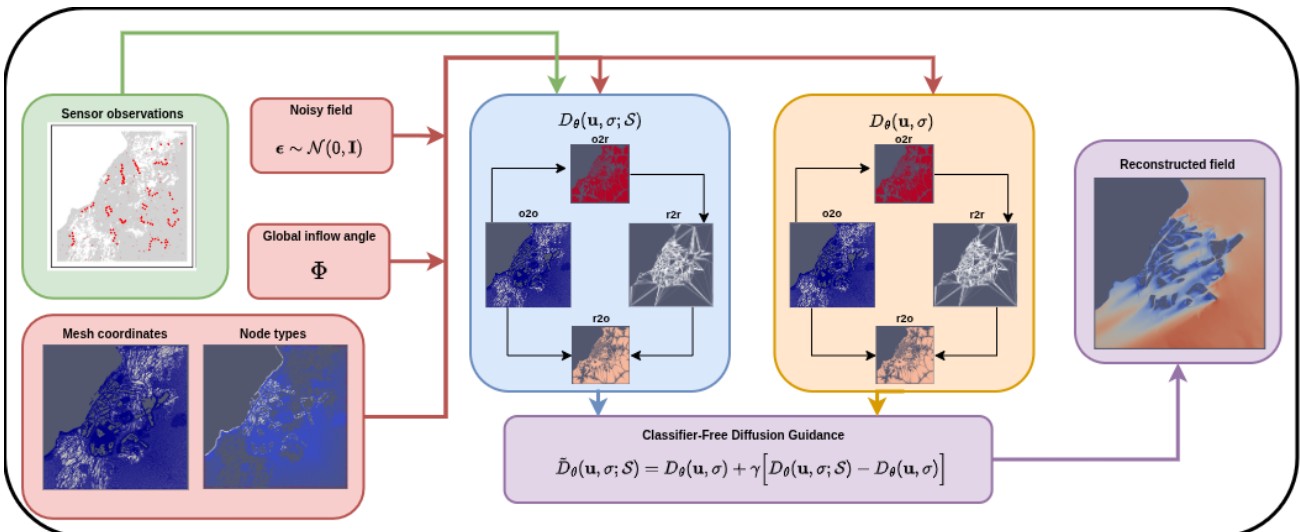

*Figure 4.* Generative data assimilation with classifier-free guidance. At each diffusion step, an unconditional denoiser estimates the geometry-conditioned prior, while a sensor-conditioned denoiser incorporates measurement information. Their difference provides an observation-driven correction, and the guidance weight $\gamma$ controls the strength of this correction.

**Implications for urban flow assimilation.** Embedding this tempered posterior sampling mechanism within a multiscale graph diffusion model enables data assimilation without explicit linearization, adjoint solvers, or fixed reduced subspaces. The reverse diffusion process progressively transforms Gaussian noise into flow fields that are globally coherent, locally accurate near obstacles, and consistent with sparse measurements. As a result, GenDA supports robust assimilation across unseen geometries, sensor layouts, and observation densities within a single, unified generative framework.

## 3. Results

This section evaluates GenDA in terms of reconstruction accuracy and robustness to realistic sensing configurations on the two held-out altitude slices, which induce unseen mesh geometries within the studied urban-flow distribution. Unless otherwise stated, results are aggregated over all wind directions, both test slices, and multiple random sensor realizations. We compare against three complementary classes of baselines: (i) two deterministic supervised GNN baselines, MeshGraphNet (MGN) (Pfaff et al., 2021) and its multiscale variant (MS-MGN) (Fortunato et al., 2022); (ii) two reduced-order data-assimilation baselines, Low-Cost SVD (LCSVD) (Hetherington & Le Clainche, 2025) and POD-3DVar (Vermeulen & Heemink, 2006); and (iii) a stochastic diffusion baseline, MultiMesh-VP (Song et al., 2021), which uses the same multiscale graph backbone as GenDA but replaces the unconditional/conditional classifier-free guidance formulation with direct observation conditioning. Beyond the main comparisons reported here, we include additional analyses in the appendix: a detailed sweep over observation counts, ablations of classifier-free guidance (CFG) and the multiscale graph hierarchy, and qualitative results under different sensing configurations (Appendix C).

### 3.1. Comparison to learned and non-learned baselines

We report three complementary metrics throughout this section. Reconstruction accuracy is measured using the Relative Root-Mean-Square Error (RRMSE), which quantifies the $\ell_2$ error of the reconstructed velocity field normalized by the total energy of the reference flow. Directional agreement is evaluated using the mean cosine similarity (also known as the Modal Assurance Criterion, MAC), which measures alignment between predicted and reference velocity vectors independently of magnitude. Finally, spatial organization is assessed using the Structural Similarity Index (SSIM), computed on interpolated velocity-magnitude fields, which captures the preservation of coherent flow structures such as wakes, shear layers, and recirculation zones. Formal metric definitions are provided in Appendix B.3.

Table 1 summarizes reconstruction performance at three representative sensor densities. GenDA consistently achieves the lowest RRMSE and the highest directional agreement across all densities, while also providing the best or near-best structural similarity. The comparison spans deterministic graph regression baselines, reduced-order DA baselines, and a stochastic diffusion baseline using the same multiscale backbone, allowing us to separate the effects of graph architecture, reduced-order assimilation, diffusion modeling, and classifier-free posterior guidance.

In the sparse regime (300 observations out of $\sim 3 \times$

*Table 1.* Reconstruction performance averaged over all test scenarios. Lower RRMSE (↓) and higher SSIM / cosine similarity (MAC) (↑) indicate better agreement with the reference flow.

| Model | RRMSE (↓) | SSIM (↑) | MAC (↑) |
|---|---|---|---|
| **300 Obs.** | | | |
| MGN | $0.426 \pm 0.138$ | $0.678 \pm 0.036$ | $0.885 \pm 0.120$ |
| MS-MGN | $0.412 \pm 0.131$ | $0.655 \pm 0.032$ | $0.899 \pm 0.101$ |
| LCSVD | $0.360 \pm 0.127$ | $0.831 \pm 0.052$ | $0.915 \pm 0.090$ |
| POD-3DVar | $0.375 \pm 0.168$ | $0.814 \pm 0.088$ | $0.909 \pm 0.101$ |
| MultiMesh-VP | $0.415 \pm 0.124$ | $0.706 \pm 0.041$ | $0.907 \pm 0.083$ |
| GenDA (Ours) | $\mathbf{0.310 \pm 0.095}$ | $\mathbf{0.834 \pm 0.029}$ | $\mathbf{0.938 \pm 0.066}$ |
| **3000 Obs.** | | | |
| MGN | $0.382 \pm 0.129$ | $0.642 \pm 0.037$ | $0.905 \pm 0.101$ |
| MS-MGN | $0.353 \pm 0.128$ | $0.669 \pm 0.033$ | $0.927 \pm 0.092$ |
| LCSVD | $0.354 \pm 0.124$ | $0.834 \pm 0.050$ | $0.918 \pm 0.088$ |
| POD-3DVar | $0.368 \pm 0.165$ | $0.819 \pm 0.088$ | $0.913 \pm 0.098$ |
| MultiMesh-VP | $0.258 \pm 0.076$ | $0.777 \pm 0.032$ | $0.962 \pm 0.031$ |
| GenDA (Ours) | $\mathbf{0.190 \pm 0.054}$ | $\mathbf{0.877 \pm 0.020}$ | $\mathbf{0.977 \pm 0.023}$ |
| **8000 Obs.** | | | |
| MGN | $0.318 \pm 0.111$ | $0.633 \pm 0.034$ | $0.932 \pm 0.078$ |
| MS-MGN | $0.277 \pm 0.094$ | $0.681 \pm 0.028$ | $0.954 \pm 0.052$ |
| LCSVD | $0.354 \pm 0.124$ | $0.834 \pm 0.050$ | $0.918 \pm 0.087$ |
| POD-3DVar | $0.367 \pm 0.164$ | $0.820 \pm 0.087$ | $0.914 \pm 0.098$ |
| MultiMesh-VP | $0.189 \pm 0.056$ | $0.822 \pm 0.025$ | $0.978 \pm 0.018$ |
| GenDA (Ours) | $\mathbf{0.136 \pm 0.042}$ | $\mathbf{0.908 \pm 0.017}$ | $\mathbf{0.988 \pm 0.011}$ |

$10^5$ nodes, $\approx 0.1\%$ coverage), GenDA reduces RRMSE from 0.426/0.412 for MGN/MS-MGN, 0.360/0.375 for LCSVD/POD-3DVar, and 0.415 for MultiMesh-VP to 0.310. At 8000 observations, GenDA reaches 0.136 RRMSE and 0.988 MAC, compared with 0.189 RRMSE and 0.978 MAC for MultiMesh-VP, the strongest non-GenDA learned generative baseline. These results indicate that the improvement is not explained solely by the multiscale graph backbone, reduced-order assimilation, or diffusion modeling alone. Rather, the combination of a learned geometry-aware prior with sensor-conditioned classifier-free guidance provides a more effective posterior reconstruction mechanism on large unstructured urban meshes.

Figure 5 provides representative qualitative reconstructions of the velocity magnitude $|\mathbf{u}|$ across multiple wind directions and sensor layouts, with observation counts shown above each column. The supervised GNN baselines (MGN, MS-MGN) capture the coarse flow organization but tend to oversmooth wakes and smear localized recirculation structures, especially in regions far from sensors. The reduced-order DA baselines (LCSVD and POD-3DVar) improve some large-scale features but remain limited near sharp gradients, obstacle-induced wakes, and complex channeling regions. MultiMesh-VP, which uses the same multiscale

graph backbone within a direct observation-conditioned diffusion model, produces sharper reconstructions than the deterministic GNN baselines but still exhibits larger localized errors and less consistent wake placement than GenDA. GenDA best preserves obstacle-induced organization, including wake extent, recirculation patterns, and channeling, with residual errors mainly concentrated along narrow shear layers and wake boundaries.

### 3.2. Robustness to sensor sampling strategy

Real deployments rarely match uniform random placement: sensors may form localized clusters (dense monitoring zones) or follow mobile trajectories (e.g., vehicle-mounted measurements). To evaluate robustness under distribution shift in the observation operator, we compare GenDA under three matched-count sampling strategies: *random* (uniformly sampled nodes), *cloud* (compact neighborhoods around seed nodes), and *trajectory* (path-like connected routes). Quantitative results are reported in Table 2 for 100-4000 observations (∼0.03-1.3% coverage).

Performance is highest for random sampling, which matches the training distribution, but GenDA remains robust under clustered and trajectory-based observations with moderate degradation. This indicates that the learned prior contributes

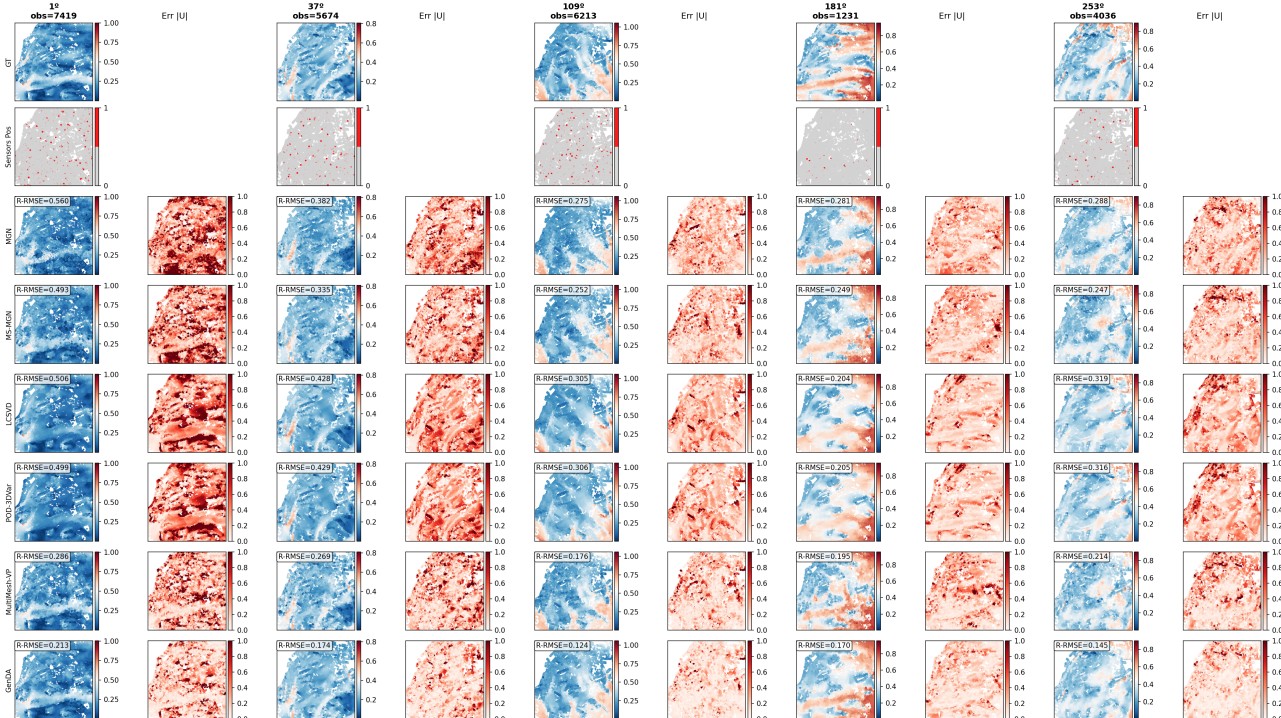

*Figure 5.* Reconstruction of velocity magnitude |**u**| for multiple wind directions (angles and sensor layouts selected as representative examples from the test set; observation counts are shown above each column). Top row: ground truth |**u**|. Second row: sensor locations. Rows 3-8: reconstructions from MGN, MS-MGN, LCSVD, POD-3DVar, MultiMesh-VP and GenDA. For each wind direction, the left panel shows the reconstructed |**u**| field and the right panel shows the corresponding pointwise relative error, highlighting spatial regions where each method deviates from the ground truth.

*Table 2.* Effect of sensor sampling strategy on reconstruction performance (100-4000 sensors, ∼0.03-1.3% coverage). Values are mean ± standard deviation over all angles, test slices and multiple sensor placements runs.

| Strategy | RRMSE (↓) | SSIM (↑) | MAC (↑) |
|---|---|---|---|
| Random | $0.291 \pm 0.084$ | $0.840 \pm 0.025$ | $0.947 \pm 0.051$ |
| Cloud | $0.354 \pm 0.109$ | $0.824 \pm 0.032$ | $0.918 \pm 0.087$ |
| Trajectory | $0.345 \pm 0.102$ | $0.825 \pm 0.031$ | $0.923 \pm 0.081$ |

meaningful global structure even when observations are spatially biased or constrained. Additional qualitative comparisons for these strategies, are provided in Appendix C.

### 3.3. Held-out sensor interpolation

To directly evaluate interpolation at unseen measurement locations, we perform a held-out sensor experiment. For each test case, we first sample a fixed sensor network and then withhold a subset of its measurements during inference. The model is evaluated only at the withheld sensor nodes, rather than over the full mesh. This diagnostic tests whether the assimilated field can accurately predict measurements that belong to the sensing network but are intentionally hidden from the model.

Table 3 reports held-out sensor RRMSE and directional

agreement. GenDA achieves the lowest held-out RRMSE and the highest MAC across all observation densities. In the sparse 300-observation setting, GenDA reduces held-out RRMSE from $0.444/0.436$ for MGN/MS-MGN, $0.346/0.341$ for LCSVD/POD-3DVar, and $0.447$ for MultiMesh-VP to $0.318$. At 8000 observations, GenDA reaches $0.160$ RRMSE and $0.982$ MAC, compared with $0.222$ RRMSE and $0.969$ MAC for MultiMesh-VP, the strongest non-GenDA generative baseline. These results confirm that the improvements observed in full-field reconstruction also translate to prediction accuracy at withheld measurement locations.

Additional quantitative and qualitative analyses are reported in Appendix C. These include extended reconstructions across observation densities, uncertainty-aware ensemble evaluation, a single-mesh ablation of the multiscale graph

*Table 3.* Held-out sensor interpolation. A fixed sensor network is sampled, a subset of measurements is withheld during inference, and error is evaluated only on the withheld sensor nodes. Values are mean $\pm$ standard deviation over 60 evaluations.

| Held-out RRMSE ($\downarrow$) | | | |
| --- | --- | --- | --- |
| Model | 300 | 3000 | 8000 |
| MGN | $0.444 \pm 0.155$ | $0.413 \pm 0.146$ | $0.359 \pm 0.131$ |
| MS-MGN | $0.436 \pm 0.152$ | $0.363 \pm 0.121$ | $0.309 \pm 0.099$ |
| LCSVD | $0.346 \pm 0.131$ | $0.334 \pm 0.125$ | $0.333 \pm 0.121$ |
| POD-3DVar | $0.341 \pm 0.123$ | $0.330 \pm 0.119$ | $0.329 \pm 0.116$ |
| MultiMesh-VP | $0.447 \pm 0.143$ | $0.291 \pm 0.091$ | $0.222 \pm 0.069$ |
| GenDA | $\mathbf{0.318 \pm 0.101}$ | $\mathbf{0.217 \pm 0.068}$ | $\mathbf{0.160 \pm 0.049}$ |

| Held-out MAC ($\uparrow$) | | | |
| --- | --- | --- | --- |
| Model | 300 | 3000 | 8000 |
| MGN | $0.880 \pm 0.120$ | $0.885 \pm 0.125$ | $0.911 \pm 0.097$ |
| MS-MGN | $0.886 \pm 0.112$ | $0.917 \pm 0.084$ | $0.941 \pm 0.059$ |
| LCSVD | $0.921 \pm 0.091$ | $0.922 \pm 0.088$ | $0.924 \pm 0.085$ |
| POD-3DVar | $0.924 \pm 0.085$ | $0.924 \pm 0.085$ | $0.926 \pm 0.081$ |
| MultiMesh-VP | $0.891 \pm 0.101$ | $0.949 \pm 0.044$ | $0.969 \pm 0.027$ |
| GenDA | $\mathbf{0.939 \pm 0.064}$ | $\mathbf{0.968 \pm 0.032}$ | $\mathbf{0.982 \pm 0.016}$ |

hierarchy, and a dedicated sweep of the classifier-free guidance (CFG) weight $\gamma$. The guidance ablation compares unconditional sampling ($\gamma=0$), purely conditional denoising ($\gamma=1$), and larger guidance values, showing that intermediate guidance provides the most robust balance between the learned geometry-aware prior and measurement consistency. These analyses further support the interpretation of CFG as a controllable posterior-sampling mechanism, while the uncertainty evaluation shows that GenDA's generated ensembles remain informative under sparse observations.

## 4. Conclusions

We presented GenDA, a generative diffusion-based data assimilation framework for reconstructing urban wind fields from sparse and irregular sensor observations on unstructured meshes. The model samples flow fields conditioned on partial observations and generalizes across held-out altitude slices, wind directions, observation densities, and sensor configurations within the studied urban-flow distribution, without requiring retraining or fine-tuning at test time.

Compared to supervised graph baselines, reduced-order data assimilation baselines, and a stochastic diffusion baseline, GenDA consistently produces lower reconstruction error and higher structural and directional agreement with the ground truth, especially under severe observation sparsity. Even with as few as 300 sensors ($\sim$0.1% coverage), GenDA reconstructs coherent wake structures and recirculation zones that competing methods struggle to recover. Unlike deterministic predictors, the proposed approach performs generative posterior reconstruction, producing ensembles of physically plausible flow fields that support uncertainty-aware inference.

We also evaluated the model's robustness to sensor placement strategy. Despite being trained only with randomly sampled sensors, GenDA generalizes well to clustered and trajectory-based patterns. Although performance is highest under random sampling, mirroring the training distribution, the degradation for structured sensors remains moderate. These results indicate that the learned geometry-aware prior enables effective assimilation under heterogeneous and non-ideal sensing configurations.

The present empirical validation is limited to steady two-dimensional slices extracted from RANS simulations of a single urban neighborhood. Broader transfer to other PDE systems, temporal forecasting, and full three-dimensional data assimilation remain important directions for future work. The present model learns physical structure implicitly from the CFD training distribution and does not yet enforce PDE residuals, mass conservation, or boundary conditions as hard constraints. Future directions include incorporating temporal dynamics to support real-time prediction with streaming sensors, jointly learning across altitudes for volumetric flow completion, and integrating physics-based constraints or solvers within the generative model to further enhance physical consistency (Barragán et al., 2025). Combining GenDA with active sensor selection or learning-based placement strategies (Vishwasrao et al., 2025) could further enable closed-loop monitoring and adaptive sensing in complex environments.

GenDA provides a flexible and scalable approach to generative data assimilation on unstructured meshes, bridging diffusion-based generative modeling, graph-based learning, and classical data assimilation concepts. More broadly, the results suggest that learned geometry-aware priors and guided diffusion sampling are a promising direction for sparse-observation inverse problems in complex, geometry-dependent domains.

## Acknowledgments

This work was supported by the Industrial PhD Program of the 'Comunidad de Madrid' under project reference IND2024/TIC-34540. The authors acknowledge the MODELAIR project that has received funding from the European Union's Horizon Europe research and innovation programme under the Marie Sklodowska-Curie grant agreement No. 101072559. The results of this publication reflect only the author's views and do not necessarily reflect those of the European Union. The European Union can not be held responsible for them. S.L.C. acknoledges the grant PID2023-147790OB-I00 funded by MCIU/AEI/10.13039 /501100011033 /FEDER, UE. The authors gratefully acknowledge the Universidad Politécnica de Madrid (www.upm.es) for providing computing resources on the Magerit Supercomputer.

## Impact Statement

This work aims to improve reconstruction of urban wind fields from sparse sensor observations, with potential benefits for air-quality monitoring, pedestrian comfort assessment, heat-risk analysis, pollutant dispersion estimation, and urban planning. By reducing the need for dense sensor deployments or repeated high-cost CFD simulations, methods such as GenDA could support faster environmental assessment in complex urban areas. However, deployment in real cities would require careful validation against physical measurements, calibration under sensor noise and bias, and assessment across diverse urban layouts and weather conditions. Unequal sensor coverage across neighborhoods could also lead to uneven reconstruction quality, which should be considered when using such systems for public decision-making. The method should therefore be viewed as a decision-support tool rather than a replacement for validated physical modeling, expert assessment, or regulatory monitoring.

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

# Appendix

## A. Urban flows dataset and simulation details

The dataset is derived from high-resolution CFD simulations of a real urban neighborhood in Bristol, UK. The domain features realistic topography and detailed building geometries, discretized into approximately 40 million unstructured cells within a cylindrical control volume. The flow regime is highly turbulent: with a maximum building height of $H_{\max} = 98\,\text{m}$ and a reference inflow velocity of $U_{\text{ref}} = 3.18\,\text{m/s}$ at $10\,\text{m}$ height, the characteristic Reynolds number is approximately $\text{Re} \approx 2.08 \times 10^7$. Further details on the base simulation database are available at modelair.eu.

To construct a 2D dataset suitable for our graph-based diffusion model, we extract horizontal slices from the 3D domain at six altitudes ($z = \{15, 20, 28, 35, 40, 45\}$ m). Each slice intersects the 3D geometry differently, producing a unique 2D unstructured mesh that captures specific building cross-sections and terrain variations at that height. The domain is cropped to a $[-1000, 1000] \times [-1000, 1000]$ m region centered on the urban core.

Figure 6 illustrates the variability of the dataset. It displays the mesh topology for both training and test slices, highlighting how the obstacle configuration changes with altitude. Additionally, it visualizes sample flow fields for different inflow wind directions, demonstrating the diversity of wake structures and channeling effects the model must learn to reconstruct.

Each 2D mesh includes node-type annotations distinguishing between fluid, wall, and boundary nodes, which are used as categorical conditioning inputs during training. For each altitude slice, the dataset contains 360 stationary flow fields, each corresponding to a distinct inflow wind direction. The fields are represented as $(U_x, U_y)$ velocity components per node. We reserve four slices for training ($z = \{15, 20, 28, 45\}$ m) and hold out two slices ($z = \{35, 40\}$ m) for testing, ensuring that the model is evaluated on geometries not seen during optimization.

### A.1. Mesh Decimation

To generate the reduced meshes required for the multiscale architecture, we apply the geometry-aware decimation algorithm of Schroeder et al. (1992). The target reduction factor is approximately $5\times$. The decimation process is configured to preferentially retain vertices near building boundaries and regions of high geometric curvature, while applying stronger downsampling in open, slowly varying regions. This ensures that the reduced graph $\mathcal{G}^r$ preserves the crucial obstacle footprints and domain boundaries of the original mesh $\mathcal{G}^o$. Table 4 summarizes the complexity of these graph structures.

*Table 4.* Graph statistics for the hierarchical mesh architecture across all altitude slices. The "Original" nodes represent the resolution at which predictions and observations are defined, while "Reduced" nodes form the latent graph for efficient message passing.

| Slice | Orig. nodes | Red. nodes | o2o edges | o2r edges | r2r edges | r2o edges |
|---|---|---|---|---|---|---|
| $z_{15}$ | 302.011 | 62.929 | 1.686.618 | 239.082 | 298.054 | 239.082 |
| $z_{20}$ | 302.043 | 54.305 | 1.674.914 | 247.738 | 278.770 | 247.738 |
| $z_{28}$ | 301.532 | 56.889 | 1.689.602 | 244.643 | 286.802 | 244.643 |
| $z_{35}$ | 310.655 | 56.319 | 1.777.546 | 254.336 | 291.246 | 254.336 |
| $z_{40}$ | 305.735 | 58.222 | 1.783.482 | 247.513 | 300.044 | 247.513 |
| $z_{45}$ | 300.780 | 55.557 | 1.766.300 | 245.223 | 296.916 | 245.223 |

## B. Additional Model and Training Details

### B.1. Background: score-based diffusion

Diffusion models define a generative process by progressively denoising samples starting from a Gaussian prior toward a target data distribution. In practice, both training and inference operate over multiple noise levels: a forward process corrupts the data with increasing noise, and a reverse process iteratively removes noise over $T$ denoising steps following a predefined noise schedule $\{\sigma_t\}_{t=1}^{T}$. Given a clean urban flow sample $\mathbf{u}_0 \in \mathbb{R}^{N \times C}$ on a mesh with $N$ nodes and $C$ velocity components, a noisy version is constructed as

$$\mathbf{u} = \mathbf{u}_0 + \sigma\boldsymbol{\epsilon}, \quad \boldsymbol{\epsilon} \sim \mathcal{N}(0, \mathbf{I}), \tag{6}$$

where $\sigma$ is drawn from a predefined noise schedule. The task of the denoiser is to predict the clean field $\mathbf{u}_0$ from the noisy input $(\mathbf{u}, \sigma)$, implicitly learning the score function $\nabla_{\mathbf{u}} \log p(\mathbf{u}; \sigma)$ associated with the noise-dependent data distribution.

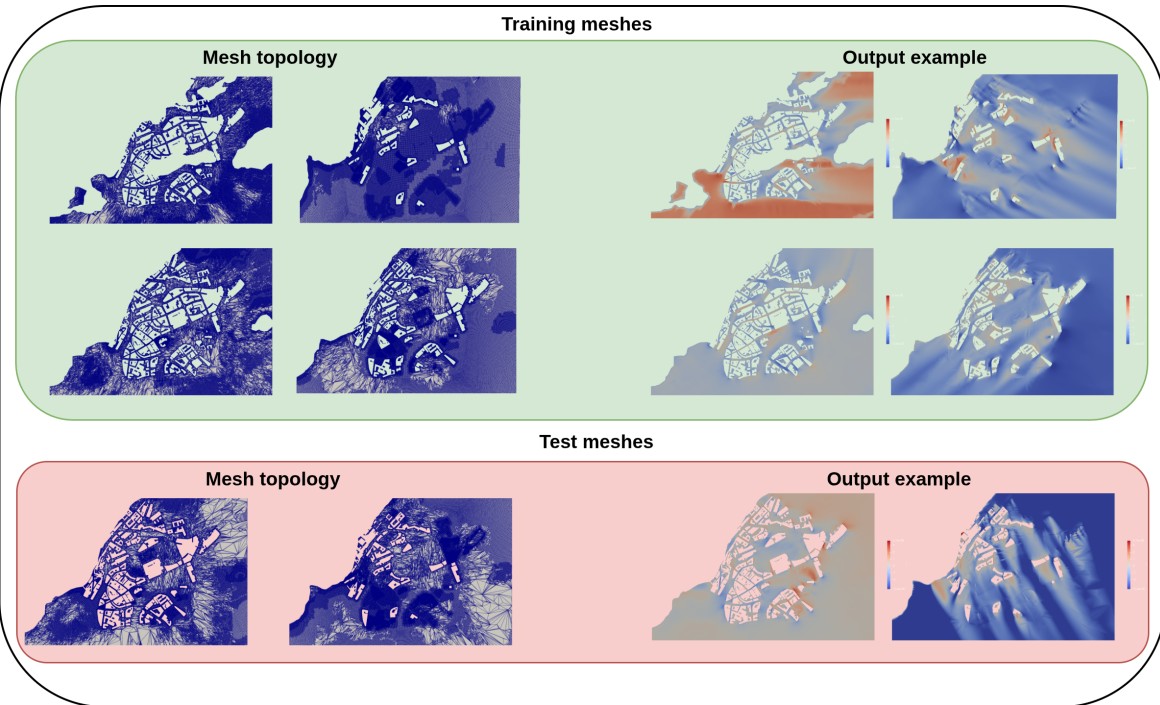

*Figure 6.* Training and evaluation meshes differ in both obstacle configuration and mesh resolution. The model must adapt to new mesh topologies after training. On the right, we observe examples of flow patterns change with the inflow wind direction; each mesh includes 360 possible inflow angles.

To stabilize learning across noise levels, we adopt the Elucidated Diffusion Model (EDM) preconditioned formulation of (Karras et al., 2022), where the denoiser $D_\theta$ produces a corrected estimate of $\mathbf{u}_0$ through

$$D_\theta(\mathbf{u}, \sigma) = c_{\text{skip}}(\sigma)\mathbf{u} + c_{\text{out}}(\sigma)F_\theta(c_{\text{in}}(\sigma)\mathbf{u},\ c_{\text{noise}}(\sigma))\,, \tag{7}$$

with $(c_{\text{in}}, c_{\text{out}}, c_{\text{skip}})$ determined by the noise schedule and $F_\theta$ a neural network mapping to learned feature spaces.

The denoiser is trained by minimizing a noise-weighted mean-squared error:

$$\mathcal{L} = \mathbb{E}_{\mathbf{u}_0, \boldsymbol{\epsilon}, \sigma} \left[ \lambda(\sigma)\ \|D_\theta(\mathbf{u}_0 + \sigma\boldsymbol{\epsilon}, \sigma) - \mathbf{u}_0\|_2^2 \right]\,, \tag{8}$$

where $\lambda(\sigma)$ balances contributions across noise levels. After training, sampling proceeds by initializing $\mathbf{u}_T$ at high noise and integrating the reverse denoising process across decreasing $\sigma$.

To operate on unstructured domains, $F_\theta$ is implemented as a multiscale graph neural network (GNN), enabling message passing on both the original and a coarsened version of each mesh. This architecture is inspired by MeshGraphNets (Pfaff et al., 2021) and Multiscale MeshGraphNets (Fortunato et al., 2022).

Full hyperparameters of the diffusion process are given in Appendix B.3.

### B.2. Graph neural network message passing and conditioning

We summarize the message-passing and conditioning equations used in the denoiser described in Section 2. The notation follows that of MultiScale MeshGraphNets (Fortunato et al., 2022), simplified to two mesh levels: the original mesh (index $o$) and the reduced mesh (index $r$). Each of the four subnetworks (o2o, o2r, r2r, r2o) follows an *encode–process–decode* structure similar to MeshGraphNets (Pfaff et al., 2021).

Each mesh level $l \in \{o, r\}$ defines a graph $\mathcal{G}^l = (\mathcal{V}^l, \mathcal{E}^l)$, where $\mathcal{V}^l$ is the set of nodes and $\mathcal{E}^l$ the set of directed edges. Each node $i \in \mathcal{V}^l$ carries a feature vector $\mathbf{h}_i^l \in \mathbb{R}^{d_l}$, and each edge $(i, j) \in \mathcal{E}^l$ has an associated feature vector $\mathbf{e}_{ij}^l$. For cross-level mappings (o2r or r2o), we additionally consider heterogeneous graphs $\mathcal{G}^{lr} = (\mathcal{V}^l, \mathcal{V}^r, \mathcal{E}^{lr})$ with edges $i \to j$ and features

$\mathbf{e}_{ij}^{lr}$ constructed from relative node positions and node-type identifiers. Edge features consist of the relative displacement vectors $\mathbf{d}_{ij} = (dx, dy)$ and their Euclidean norm $d_{ij} = \|\mathbf{d}_{ij}\|$, which provide the network with local geometric context invariant to absolute position.

Before message passing, the raw node inputs, comprising the noisy velocity field $\mathbf{u}_i$, node type encodings, and the sensor observation pairs $(m_i, y_i)$, and the raw edge inputs $\mathbf{f}_{ij}$ are mapped into latent space through encoder MLPs:

$$\mathbf{h}_i^{(0)} = \phi_{\text{enc}}^n([\mathbf{u}_i, \mathbf{x}_i, \text{type}_i, m_i, y_i]), \tag{9}$$

$$\mathbf{e}_{ij}^{(0)} = \phi_{\text{enc}}^e(\mathbf{f}_{ij}), \tag{10}$$

where $\mathbf{h}_i^{(0)}$ and $\mathbf{e}_{ij}^{(0)}$ serve as the initial node and edge embeddings. To support Classifier-Free Guidance, the unconditional score is obtained by evaluating the network with a null sensor mask, setting $m_i = 0$ and $y_i = 0$ for all nodes, which signals the network to ignore observation inputs.

Each subnetwork performs $K$ message-passing iterations. At iteration $k$, node embeddings $\mathbf{h}_i^{(k)}$ and edge embeddings $\mathbf{e}_{ij}^{(k)}$ are updated through learned message functions. For each edge $(i, j)$, we compute a message

$$\mathbf{m}_{ij} = \phi_e\left(\mathbf{h}_i^{(k)}, \mathbf{h}_j^{(k)}, \mathbf{e}_{ij}^{(k)}, \mathbf{g}\right), \tag{11}$$

where $\mathbf{g}$ is a global conditioning vector formed by encoding the diffusion noise level $\sigma$ and the wind direction $\Phi$. Incoming messages are aggregated per node using a permutation-invariant operator,

$$\mathbf{m}_i = \sum_{j:(i,j)\in\mathcal{E}} \rho_e(\mathbf{m}_{ij}), \tag{12}$$

and the node state is updated as

$$\mathbf{h}_i^{(k+1)} = \phi_h\left(\mathbf{h}_i^{(k)}, \mathbf{m}_i, \mathbf{g}\right). \tag{13}$$

Here, $\phi_e$ and $\phi_h$ denote MLPs applied at the edge and node levels, respectively, and $\rho_e(\cdot)$ denotes message aggregation by summation.

The global conditioning vector $g$ modulates the normalization layers in all processors (Chen et al., 2021), following the norm-conditioning scheme of GenCast (Price et al., 2024). It is implemented through an affine normalization transform:

$$\text{LayerNorm}(\mathbf{h}_i; \mathbf{g}) = \gamma(\mathbf{g})\frac{\mathbf{h}_i - \mu(\mathbf{h})}{\sigma(\mathbf{h})} + \beta(\mathbf{g}), \tag{14}$$

where $\gamma(\mathbf{g})$ and $\beta(\mathbf{g})$ are scale and bias terms predicted by an MLP applied to $\mathbf{g}$. This mechanism injects the effect of $\sigma$ and $\Phi$ consistently across layers and scales.

After $K$ message-passing iterations, each node's latent representation can be decoded back to the target dimensionality:

$$\widehat{\mathbf{y}}_i = \phi_{\text{dec}}(\mathbf{h}_i^{(K)}), \tag{15}$$

where $\phi_{\text{dec}}$ is an MLP with residual connections. In practice, this operation is only performed in the final $\texttt{r2o}$ network, to produce the denoised velocity prediction on the original mesh.

The projection from original to reduced mesh (downsample) and the reverse mapping (upsample) are performed on heterogeneous graphs:

$$\mathbf{h}_j^{r'} = \phi_{r2r}\left(\mathbf{h}_j^r, \sum_{i:(i,j)\in\mathcal{E}^{o2r}} \rho(\phi_e^{o2r}(\mathbf{h}_i^o, \mathbf{e}_{ij}^{o2r}))\right), \tag{16}$$

$$\mathbf{h}_i^{o'} = \phi_{o2o}\left(\mathbf{h}_i^o, \sum_{j:(j,i)\in\mathcal{E}^{r2o}} \rho(\phi_e^{r2o}(\mathbf{h}_j^r, \mathbf{e}_{ji}^{r2o}))\right). \tag{17}$$

This bidirectional exchange allows coarse-scale information from the reduced mesh to influence fine-scale updates on the original mesh, improving spatial coherence and enabling efficient long-range communication.

After the last upsample step, the node embeddings on the original mesh are projected to the output channels using a linear layer applied independently at each node,

$$\widehat{\mathbf{y}}_i = W_{\text{out}}\, \mathbf{h}_i^{o'} + b_{\text{out}}, \qquad i \in \mathcal{V}^o, \tag{18}$$

which yields the predicted flow field $\widehat{\mathbf{Y}} = \{\widehat{\mathbf{y}}_i\}_{i \in \mathcal{V}^o}$ over all nodes. The output is then combined with the input according to the EDM preconditioning coefficients $(c_{\text{skip}}, c_{\text{out}})$ introduced in Section 2 (Eq. 7).

Each of the four subnetworks thus contains a full encode–process–decode pipeline, with shared design principles but distinct graph connectivity. Together, they implement a hierarchical GNN system that enables both local fine-scale corrections and long-range contextual propagation within a single denoising iteration.

## B.3. Training and evaluation details

We train the model using a continuous EDM-style noise schedule, inspired by (Price et al., 2024), parameterized by $\rho{=}7.0$:

$$\sigma(u) = \left(\sigma_{\max}^{1/\rho} + u(\sigma_{\min}^{1/\rho} - \sigma_{\max}^{1/\rho})\right)^{\rho}, \qquad u \sim \mathcal{U}(0, 1), \tag{19}$$

together with the preconditioned denoiser and loss defined in Eqs. (7)–(8). We set $\sigma_{\min} = 0.02 \cdot 0.2757/0.5$ and $\sigma_{\max} = 88.0 \cdot 0.2757/0.5$ to match the empirical velocity scale of the dataset.

At each iteration, a new random subset of sensors is generated by uniformly sampling a coverage ratio between $0.1\%$ and $5\%$ of mesh nodes and assigning their $(m_i, y_i)$ values from the ground-truth field. This randomization enforces variability in both the number and spatial placement of observations. With probability $p_{\text{uc}}{=}0.1$, all sensor inputs are dropped to train the unconditional branch.

We optimize with AdamW using cosine learning rate decay (learning rate $1{\times}10^{-4}$, $\beta_1{=}0.9$, $\beta_2{=}0.999$, weight decay $1{\times}10^{-2}$, and global-norm clipping at $5.0$). Training runs for 200,000 steps with batch size 2 per GPU on four NVIDIA A100 GPUs. All node, edge, and processor MLPs use hidden size 64, and the global conditioning vector (noise level $\sigma$ and wind direction $\phi$) is projected to 64 dimensions and injected through conditional layer normalization.

For inference, we use stochastic EDM sampling with $T{=}20$ steps (which we found is a good trade-off for computational cost and performance) and the same $\rho{=}7.0$ schedule, with $\sigma_{\min} = 0.02 \cdot 0.2757/0.5$, $\sigma_{\max} = 80.0 \cdot 0.2757/0.5$, and $(s_{\text{churn}}, s_{\min}, s_{\max}, s_{\text{noise}}) = (2.5, 0.75, \infty, 1.05)$. Classifier-free guidance uses a fixed $\gamma{=}2.0$ across all steps.

The overall training and sampling hyperparameters follow the diffusion-based flow modeling approach of GenCast (Price et al., 2024) and standard EDM settings; the main adaptation to our mesh-native formulation is the dataset-dependent velocity scaling used to set $\sigma_{\text{data}}$ (and thus $\sigma_{\min}$ and $\sigma_{\max}$), while the remaining parameters are kept fixed across experiments. The guidance weight $\gamma$ is selected empirically through preliminary experiments across representative sensor densities and wind directions, and we use $\gamma = 2.0$ in all experiments reported here. In particular, the parameter $\sigma_{\text{data}}$ is redefined using the empirical velocity magnitudes of our dataset, as we scale velocity components by their maximum values rather than by standardization. This dataset-specific rescaling improves numerical stability and reconstruction quality during training.

Model performance is evaluated directly on the original unstructured mesh using three complementary metrics. The first is the relative root-mean-square error (RRMSE), which quantifies magnitude discrepancies relative to the total energy of the reference field. For any scalar or vector quantity $q \in \{\mathbf{u}, U_x, U_y, |\mathbf{u}|\}$, where $\mathbf{u} = (U_x, U_y)$ denotes the 2D velocity vector at each mesh node, $U_x$ and $U_y$ are its Cartesian components, and $|\mathbf{u}| = \sqrt{U_x^2 + U_y^2}$ is the velocity magnitude, we compute

$$\text{RRMSE}(q) = \sqrt{\frac{\sum_{i=1}^{N}\left\|q_i^{\text{pred}} - q_i^{\text{true}}\right\|^2}{\sum_{i=1}^{N}\left\|q_i^{\text{true}}\right\|^2}}, \tag{20}$$

where the numerator measures reconstruction error and the denominator normalizes by the flow's overall energy. Since our dataset consists of steady Reynolds-averaged Navier–Stokes mean fields, all reported errors are computed on mean velocities (not on temporal fluctuations). With the normalization in Eq. (20), $\text{RRMSE}(\mathbf{u})$ is the relative $\ell_2$ error of the reconstructed velocity field on the mesh.

To evaluate directional agreement independently of magnitude, we use the mean cosine similarity between predicted and true velocity vectors:

$$\langle \cos\theta \rangle = \frac{1}{N} \sum_{i=1}^{N} \frac{\mathbf{u}_i^{\text{pred}} \cdot \mathbf{u}_i^{\text{true}}}{\left\| \mathbf{u}_i^{\text{pred}} \right\| \left\| \mathbf{u}_i^{\text{true}} \right\| + \varepsilon}, \tag{21}$$

where a small $\varepsilon$ avoids division by zero at near-stagnant nodes. Values close to one indicate strong directional alignment, corresponding to high modal coherence between predicted and true flows. This metric is also known as the Modal Assurance Criterion (MAC) when used to assess directional agreement (Mendez et al., 2021).

Finally, to assess structural fidelity, we interpolate the velocity magnitude onto a regular grid and compute the Structural Similarity Index (SSIM) (Wang et al., 2004). SSIM values lie in $[0, 1]$, with 1 indicating identical spatial structure. All metrics are computed per wind direction and reported as mean $\pm$ standard deviation across directions and available slices.

### B.4. Baseline details

We compare GenDA against deterministic graph-learning baselines, reduced-order data-assimilation baselines, and a stochastic diffusion baseline. The supervised graph baselines are MeshGraphNet (MGN) (Pfaff et al., 2021) and a multiscale variant inspired by Multiscale MeshGraphNets (MS-MGN) (Fortunato et al., 2022). MGN operates directly on the original unstructured mesh and predicts the full velocity field from the observed node features in a deterministic feed-forward manner. MS-MGN uses the same observation encoding but introduces the reduced mesh hierarchy used by GenDA, providing a deterministic baseline that isolates the effect of multiscale message passing without diffusion sampling or classifier-free guidance.

For reduced-order data assimilation, we include Low-Cost SVD (LCSVD) (Hetherington & Le Clainche, 2025) and POD-3DVar (Vermeulen & Heemink, 2006). LCSVD reconstructs the flow in a data-driven low-dimensional subspace using sensor observations and an optimal-sensor-placement-inspired reduced representation. POD-3DVar is a variational data-assimilation baseline in a POD subspace: the analysis state is obtained by minimizing a background-plus-observation cost function over POD coefficients, using observation and background covariance weights. These baselines represent classical alternatives that assimilate observations through low-dimensional linear representations rather than sampling directly in the full graph-defined state space. Finally, MultiMesh-VP is a stochastic diffusion baseline based on variance-preserving score-based diffusion (Song et al., 2021). It uses the same multiscale graph backbone as GenDA but conditions directly on observations, without the unconditional/conditional classifier-free guidance decomposition. This baseline tests whether the gains of GenDA arise from the proposed posterior-guidance formulation rather than from diffusion sampling or multiscale message passing alone.

## C. Additional Results

### C.1. Effect of classifier-free guidance strength

Classifier-free guidance (CFG) controls how strongly sensor information influences the generative reconstruction by scaling the difference between the conditional and unconditional denoiser outputs (Eq. (4) in the main text). To assess its impact, we evaluate GenDA across a range of guidance weights, from unconditional sampling ($\gamma=0$) to strongly guided sampling ($\gamma > 2$).

Figure 7 reports reconstruction metrics as a function of $\gamma$, averaged over all wind directions, test slices, and random sensor placements. When $\gamma=0$, the model ignores sensor information and samples exclusively from the learned geometry-conditioned prior. In this regime, the reconstructed fields remain physically plausible and capture large-scale organization imposed by urban geometry and inflow direction, but they exhibit large magnitude errors and poor local agreement with measurements, as reflected by high RRMSE and reduced SSIM.

At $\gamma=1$, corresponding to purely conditional denoising without explicit guidance amplification, reconstruction quality improves relative to unconditional sampling but remains consistently suboptimal. In particular, sparse-observation cases still show under-enforcement of measurement consistency, indicating that conditioning alone is insufficient to fully assimilate sensor information during the reverse diffusion process.

To make the guidance-weight selection criterion explicit, Table 5 reports an expanded sweep for representative observation densities. At 300 observations, stronger guidance improves RRMSE from 0.311 at $\gamma=1.0$ to 0.303 at $\gamma=1.5$ and 0.298 at

*Table 5.* Effect of guidance strength on RRMSE for representative observation densities. Intermediate guidance values provide the most robust trade-off across densities, while overly strong guidance becomes unstable as observational information increases.

| Obs. | $\gamma$=1.0 | $\gamma$=1.5 | $\gamma$=2.0 | $\gamma$=3.0 |
|------|--------|--------|--------|--------|
| 300  | 0.311  | 0.303  | 0.298  | **0.296** |
| 3000 | 0.193  | 0.184  | **0.183** | 0.291  |
| 8000 | 0.137  | **0.131** | 0.145  | 0.481  |

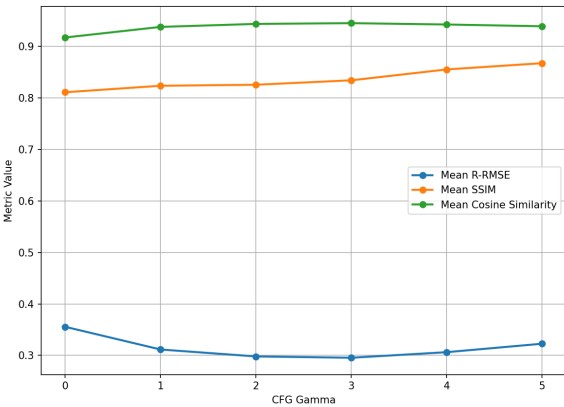

*Figure 7.* Effect of the classifier-free guidance weight $\gamma$ on reconstruction quality. Metrics are averaged over all wind directions, test slices, and random sensor placements. $\gamma$=0 corresponds to unconditional prior sampling, $\gamma$=1 to purely conditional denoising, and $\gamma > 1$ to guided posterior sampling. Optimal performance is achieved for intermediate guidance values, illustrating the balance between geometry-aware prior consistency and enforcement of measurement fidelity.

$\gamma$=2.0, with only a marginal additional gain at $\gamma$=3.0. However, this trend does not persist at higher observation densities: at 3000 observations, $\gamma$=3.0 increases RRMSE to $0.291$, and at 8000 observations it becomes clearly unstable, reaching $0.481$.

Best performance is therefore obtained for intermediate guidance values ($\gamma \in [1.5, 2.0]$), where RRMSE is minimized or remains close to the best value while MAC and SSIM stay consistently high. In this regime, the guidance term effectively acts as a learned correction toward observation-consistent states, while the unconditional branch preserves global coherence and physically plausible flow structure. For excessively large $\gamma$, performance degrades, especially at higher observation densities, suggesting that over-amplified guidance can over-correct the reverse diffusion trajectory and disrupt prior consistency.

We therefore fix $\gamma$=2 in the main experiments not because it is always the single best value for every density or metric, but because it provides a reliable global choice across observation densities, test slices, and wind directions. Overall, this behavior supports the interpretation of CFG as a controllable posterior-sampling mechanism: the unconditional diffusion model represents a learned geometry-aware prior over urban flow fields, while the guidance term injects a data-driven approximation of measurement consistency during sampling. The parameter $\gamma$ governs the balance between these two contributions, analogous to the trade-off between background and observation terms in classical data assimilation.

### C.2. Uncertainty-aware evaluation

As a generative model, GenDA produces an ensemble of plausible reconstructions rather than a single deterministic field. To evaluate whether this ensemble carries useful uncertainty information, we compute probabilistic scores under the same held-out sensor protocol used in Section 3.3. For each test case, a fixed sensor network is sampled, only 25% of the measurements are provided during inference, and uncertainty is evaluated on the remaining 75% held-out sensor nodes.

We use the continuous ranked probability score (CRPS), a proper scoring rule that compares the predictive ensemble distribution with the observed value. For an ensemble of $K$ predictions $\{q_i^{(k)}\}_{k=1}^{K}$ of a scalar quantity $q_i$ at a held-out node

*Table 6.* Held-out sensor CRPS for GenDA ensembles under the masked-sensor protocol. Lower values indicate better calibrated and sharper predictive distributions at withheld sensor locations.

| Observation setting | CRPS ($\downarrow$) |
|---|---|
| 50 | 0.058 |
| 100 | 0.055 |
| 300 | 0.048 |
| 1000 | 0.040 |

*Table 7.* Wasserstein distance between predicted and reference marginal velocity-component distributions in the 300-observation setting. Lower values indicate closer agreement with the reference distribution.

| Model | $U_x$ ($\downarrow$) | $U_y$ ($\downarrow$) |
|---|---|---|
| MGN | 0.0406 | 0.0226 |
| MS-MGN | 0.0225 | **0.0081** |
| LCSVD | 0.0197 | 0.0212 |
| POD-3DVar | 0.0185 | 0.0214 |
| MultiMesh-VP | 0.0208 | 0.0143 |
| GenDA | **0.0150** | 0.0116 |

$i$, we estimate

$$\mathrm{CRPS}_i = \frac{1}{K} \sum_{k=1}^{K} \left| q_i^{(k)} - q_i^\star \right| - \frac{1}{2K^2} \sum_{k=1}^{K} \sum_{\ell=1}^{K} \left| q_i^{(k)} - q_i^{(\ell)} \right|, \tag{22}$$

where $q_i^\star$ is the reference value. Lower CRPS indicates a predictive distribution that is both sharper and better calibrated with respect to the held-out measurement.

Table 6 reports the held-out sensor CRPS as the number of available observations increases from 50 to 1000. The score decreases monotonically, from approximately 0.058 with 50 observations to 0.040 with 1000 observations, indicating that GenDA's predictive ensembles become sharper and more accurate as more observational information is assimilated.

We also compare aggregate marginal velocity-component distributions in the sparse 300-observation setting using the Wasserstein distance between predicted and reference samples. As shown in Table 7, GenDA obtains the lowest distance for $U_x$ and remains competitive for $U_y$, indicating that its reconstructions preserve global velocity-component statistics under sparse observations.

The CRPS trend and Wasserstein comparison support the uncertainty-aware interpretation of GenDA: the model does not only improve deterministic reconstruction metrics, but also produces ensembles and marginal distributions that remain informative under sparse observations.

### C.3. Ablation of the multiscale graph hierarchy

To isolate the contribution of the multiscale graph hierarchy, we evaluate a single-mesh variant of GenDA (SM-GenDA). This model keeps the same diffusion training procedure, sensor conditioning, and classifier-free guidance mechanism, but removes the reduced mesh and cross-scale message passing. Thus, the comparison preserves the generative data-assimilation formulation while restricting information propagation to the original high-resolution mesh.

Table 8 shows that removing the reduced communication graph degrades reconstruction quality most clearly in sparse and moderate observation regimes. At 300 observations, RRMSE increases from 0.312 to 0.399, SSIM decreases from 0.823 to 0.791, and MAC decreases from 0.937 to 0.899. At 3000 observations, RRMSE increases from 0.192 to 0.254. At 8000 observations, the gap becomes small, suggesting that dense sensor coverage can partially compensate for the absence of the reduced communication graph. Overall, these results indicate that the multiscale hierarchy is most important when observations are sparse, where efficient long-range propagation is needed to distribute limited sensor information across complex urban geometry.

*Table 8.* Single-mesh GenDA ablation. SM-GenDA keeps the same diffusion and CFG-based assimilation formulation as GenDA, but removes the reduced mesh and cross-scale message passing. Values are mean $\pm$ standard deviation over test cases.

| Obs. | Model | RRMSE ($\downarrow$) | SSIM ($\uparrow$) | MAC ($\uparrow$) |
|---|---|---|---|---|
| 300 | SM-GenDA | $0.399 \pm 0.118$ | $0.791 \pm 0.034$ | $0.899 \pm 0.104$ |
| 300 | GenDA | $\mathbf{0.312 \pm 0.094}$ | $\mathbf{0.823 \pm 0.029}$ | $\mathbf{0.937 \pm 0.066}$ |
| 3000 | SM-GenDA | $0.254 \pm 0.072$ | $0.830 \pm 0.025$ | $0.959 \pm 0.042$ |
| 3000 | GenDA | $\mathbf{0.192 \pm 0.054}$ | $\mathbf{0.867 \pm 0.020}$ | $\mathbf{0.976 \pm 0.023}$ |
| 8000 | SM-GenDA | $0.142 \pm 0.042$ | $0.897 \pm 0.016$ | $0.987 \pm 0.012$ |
| 8000 | GenDA | $\mathbf{0.137 \pm 0.040}$ | $\mathbf{0.898 \pm 0.015}$ | $\mathbf{0.987 \pm 0.012}$ |

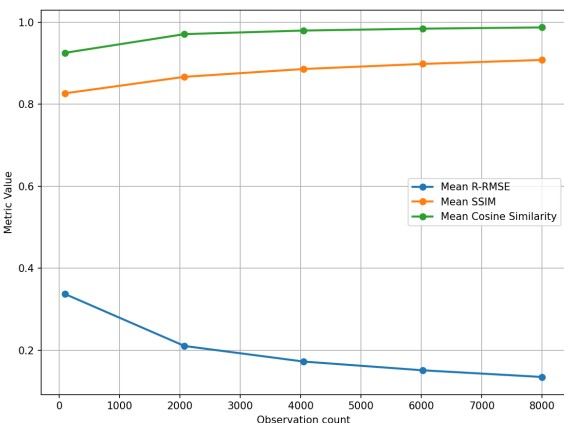

*Figure 8.* Extended effect of observation density on reconstruction quality. Metrics are averaged over all wind directions, test slices, and random sensor placements. Increasing sensor coverage yields smooth and monotonic improvements across all metrics.

### C.4. Extended analysis across observation densities

Figure 8 extends the observation-count analysis presented in the main text by reporting reconstruction metrics over a wider range of sensor densities. As the number of sensors increases, RRMSE decreases smoothly while SSIM and cosine similarity increase, indicating progressively improved recovery of both velocity magnitude and spatial organization. No abrupt transitions are observed, suggesting that GenDA degrades gracefully as observations become sparser and does not rely on a critical sensor density to function.

Figure 9 provides additional qualitative examples for a fixed wind direction while varying the number of sensors. With extremely sparse measurements, reconstructions are dominated by the learned prior and capture only the dominant flow direction and large-scale organization. As sensor coverage increases, wake structures, recirculation zones, and channeling effects emerge with correct spatial extent and intensity. At higher densities, remaining errors are localized near sharp gradients and complex wake interactions.

### C.5. Additional qualitative comparisons

Figure 10 shows a component-wise comparison of the velocity components $U_x$ and $U_y$ for a representative wind direction. These examples complement the magnitude-based visualizations in the main text and illustrate how GenDA recovers both magnitude and direction in recirculation regions and narrow street canyons, where baselines tend to exhibit directional bias and excessive smoothing.

Finally, Figure 11 provides additional qualitative comparisons across sensor sampling strategies. While random sampling yields the most accurate global reconstructions, clustered and trajectory-based measurements still allow GenDA to recover the dominant flow organization, with errors increasing primarily in regions far from observed paths or clusters.

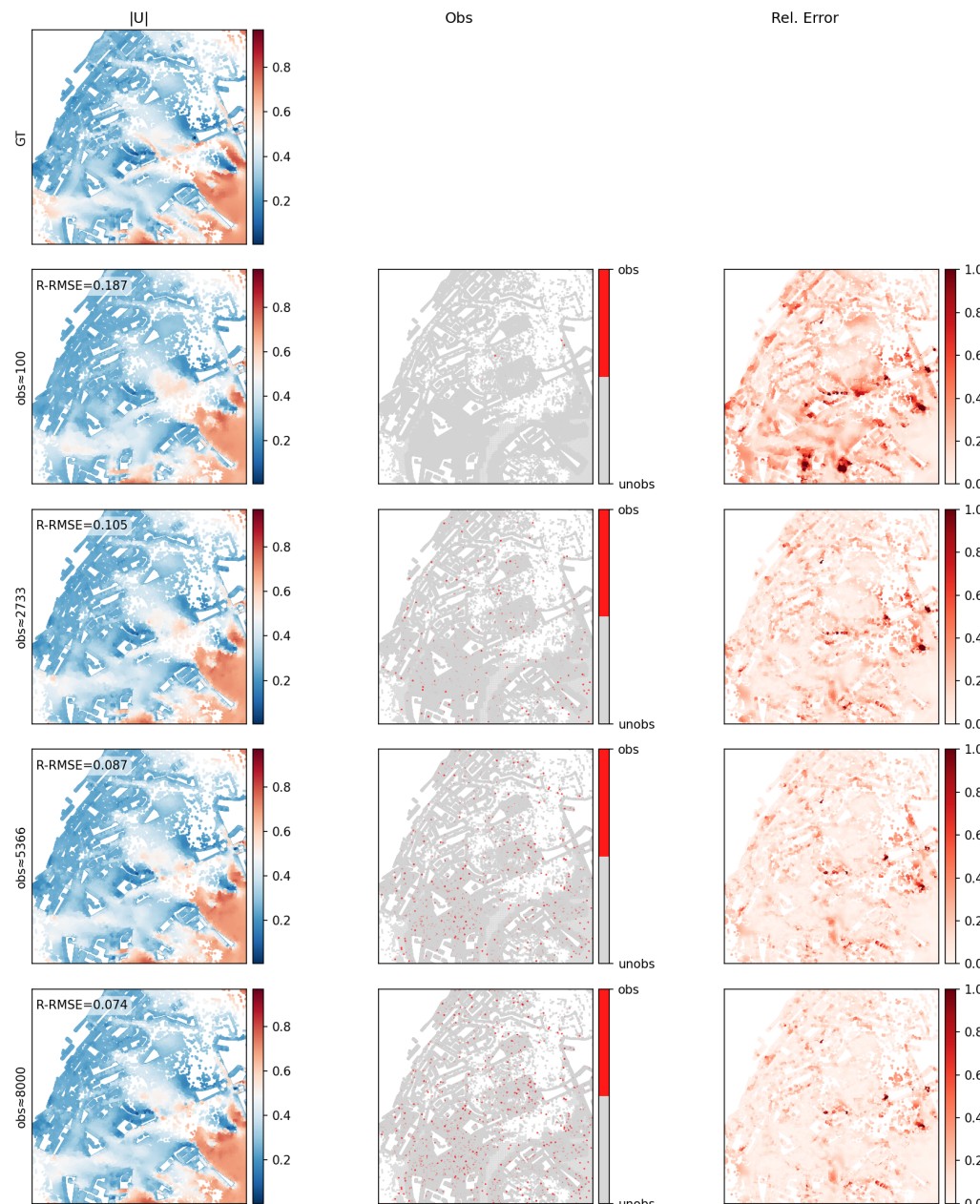

*Figure 9.* Qualitative effect of observation density on reconstructed velocity magnitude $|\mathbf{u}|$ for a representative wind direction. From top to bottom: ground truth, sensor locations, and GenDA reconstructions for increasing numbers of sensors.

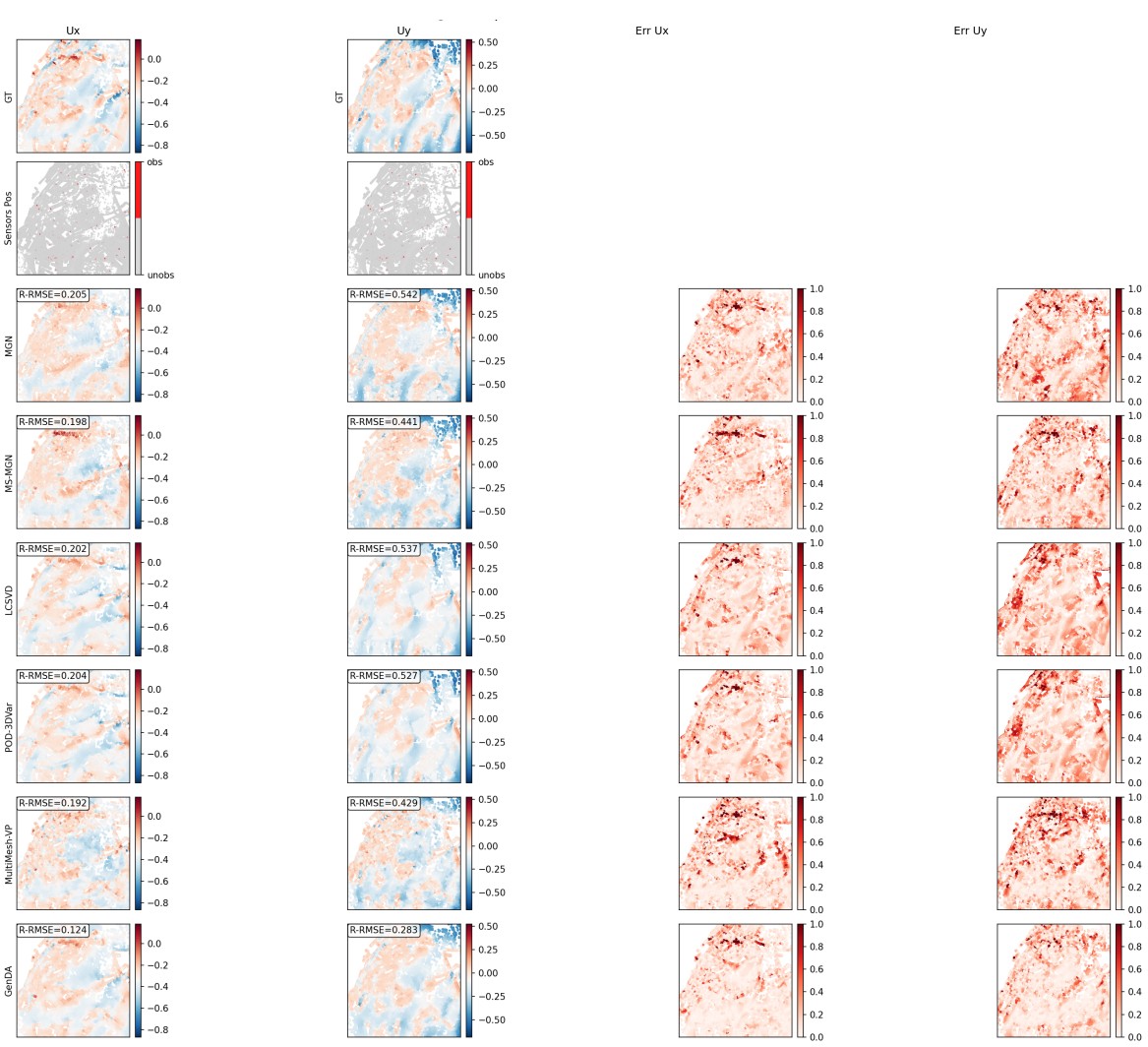

*Figure 10.* Component-wise reconstruction of velocity components $U_x$ and $U_y$ for wind direction $253°$ with 1662 observations. Top rows: ground truth and sensor locations. Bottom rows: baselines and GenDA predictions and corresponding pointwise relative error fields.

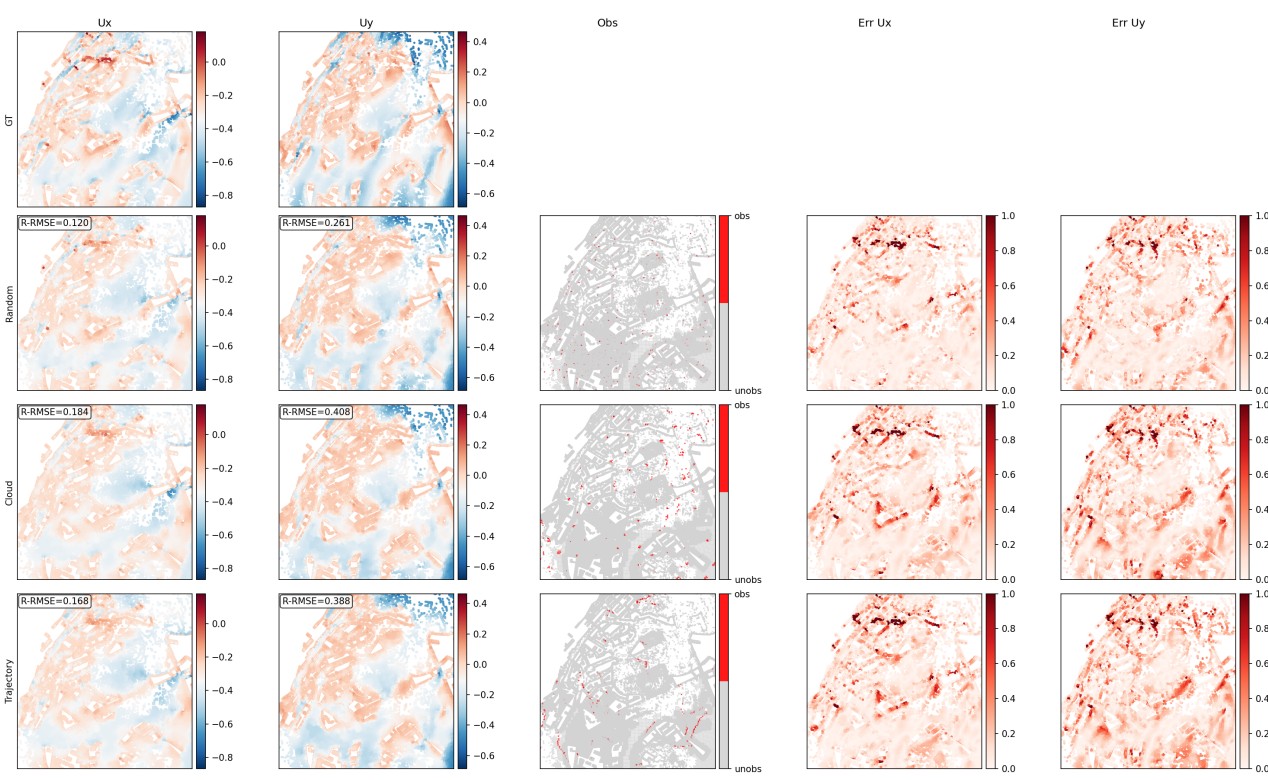

*Figure 11.* Qualitative comparison of GenDA reconstructions under different sensor sampling strategies (random, cloud, and trajectory) for wind direction $253°$ with $\sim$2177 observations.

