# OpenReview forum: "GenDA: Generative Data Assimilation on Complex Urban Areas via Classifier-Free Diffusion Guidance"
_ICML.cc/2026/Conference — ICML 2026 regular_

### Official Review · Reviewer_QgwV · 2026-03-03

**Soundness:** 1
**Presentation:** 1
**Significance:** 2
**Originality:** 2
**Overall Recommendation:** 2
**Confidence:** 4

**Summary:**

The paper uses generative data assimilation with classifier-free guidance on an irregular grid for urban wind flow reconstruction. The main contribution for the paper is to form a multiscale grid and apply generative data assimilation on the irregular grid and find it has performance improvement upon GNN baselines.

**Compliance With Llm Reviewing Policy:**

Affirmed.

**Final Justification:**

Thanks for the authors' active rebuttal. The rebuttal partially addressed my concerns. After re-reading both the rebuttal and full paper, I would like to stick to my original assessment.

**Key Questions For Authors:**

1. I am surprised that traditional methods, which utilize a prior, perform worse than the generative data assimilation methods, which do not use a prior. Can you explain why this is the case?
2. Why was the dataset split based on altitude rather than timesteps?
3. Can you explain why the baseline performance is so poor? Why is generative DA able to outperform your baseline, and what specific component of the generative DA framework contributed most to this superior performance?
4. Why did you choose to condition the model on a global wind direction? Is this a realistic experimental choice for real-world settings where global parameters may not be known a priori?

**Limitations:**

See weakness above.

**Strengths And Weaknesses:**

Strengths:
1. As far as I know, this paper is the first to apply generative data assimilation to the irregular grid.

Weaknesses:
1. The presentation is not well formalized. The training and test dataset split should be clearly stated in the main text. It would be beneficial if the authors dedicated more space to introducing the baselines and explaining the fundamental differences between generative data assimilation and traditional approaches. Additionally, the data samples in the figures require clearer captions including the altitude and time.
2. Because sensors must be located on the mesh, observations cannot be extended beyond the mesh, which limits the method’s scalability. Standard practice usually involves training an unconditional generative model and applying guidance or noise optimization for constrained generation.
3. The experimental framework is not well formalized. Traditional data assimilation (DA) methods are designed to integrate a prior with observations to produce a corrected field, rather than as a pure reconstruction task. In real-world settings where the full ground truth is unknown, it is standard to evaluate the performance using held-out stations.
4. The baselines are too weak. The GNN baseline predictions in Figure 5 appear very noisy—is the model sufficiently trained, or is there room to optimize the GNN architecture? Moreover, it would be more meaningful to compare the results against stronger traditional DA methods like 3D-Var or EnKF.  Traditional DA methods with a decent prior estimation could typically outperform generative DA, since generative DA starts purely from random noise.

---

> ### Author Rebuttal · Authors · 2026-03-31
>
> We thank the reviewer for the detailed comments and for recognizing this as the first application of generative data assimilation on irregular grids. In the revision, we will clarify the train/test splits, expand the baseline descriptions, and revise figure captions to state the altitude slice $z$, the inflow angle $\Phi$, and the steady RANS nature of the fields (which lack a time dimension).
>
> To address your concern regarding DA-style evaluation, we performed a new masked-sensor experiment: we fix a sensor configuration, withhold a subset during inference, and evaluate exclusively on the held-out sensor nodes. GenDA remains superior across all regimes, directly confirming its practical DA capabilities at unseen measurement locations:
>
> * **Held-out RRMSE (300 / 3000 / 8000 obs):** GenDA (**0.318 / 0.217 / 0.160**) vs. MGN (0.444 / 0.413 / 0.359) vs. MS-MGN (0.436 / 0.363 / 0.309) vs. LCSVD (0.346 / 0.334 / 0.333).
>
> *(Please refer to our rebuttal to **Reviewer Z6xY** for more detailed information on this experiment).*
>
> > Q1: I am surprised that traditional methods, which utilize a prior, perform worse than the generative data assimilation methods, which do not use a prior. Can you explain why?
>
> **A1:** We believe there is an important misunderstanding: GenDA *does* use a prior. The unconditional branch learns a nonlinear, geometry-aware prior over plausible flow fields directly on the graph. As stated in the manuscript, $\gamma=0$ corresponds to sampling exclusively from this learned prior. Classifier-free guidance then balances this prior with the sensor-conditioned correction.
>
> The distinction from LCSVD is not "prior vs. no prior," but the *type* of prior. LCSVD relies on a reduced linear subspace, which is less expressive when generalizing to unseen geometries with complex obstacle-induced wakes. While diffusion sampling starts from noise, the reverse process is entirely driven by our learned prior. We will make this distinction much more explicit.
>
> > Q2: Why was the dataset split based on altitude rather than timesteps?
>
> **A2:** Because the dataset consists of steady RANS simulations, there is no time axis. We chose the altitude split to probe the most relevant generalization question: can the model transfer to unseen geometries? Each altitude slice represents a distinct 2D cross-section of the 3D urban domain with new obstacle footprints and fluid-solid interactions. This forces the model to generalize to new graph structures rather than interpolate over repeated samples of the same geometry. We will move this rationale into the main text.
>
> > Q3: Can you explain why the baseline performance is so poor? Why is generative DA able to outperform your baseline, and what specific component contributed most?
>
> **A3:** The single-scale baselines operate on massive original meshes (~3×10^5 nodes, 1.7×10^6 edges), severely restricting the feasible number of message-passing layers due to memory limits. To isolate GenDA's performance gains, we ran two new ablations:
>
> * **Single-mesh GenDA:** Removing the reduced mesh degrades performance, proving the multiscale hierarchy is computationally vital for long-range communication.
> * **MultiMesh-VP:** A standard VP diffusion setup using our multiscale backbone. At 8000 observations, GenDA achieves 0.137 RRMSE while MultiMesh-VP achieves 0.189.
>
> This proves the gain stems from our specific full formulation (a learned geometry-aware prior + a sensor-conditioned branch + classifier-free guidance), not just weak baselines or simply "using diffusion."
>
> Regarding stronger classical DA methods (3D-Var or EnKF): these require dynamical forecasts and background covariance models, whereas our benchmark is a static snapshot reconstruction from steady RANS slices. LCSVD serves as the appropriate reduced-order DA comparator under these data assumptions.
>
> > Q4: Why condition the model on a global wind direction? Is this a realistic choice for real-world settings?
>
> **A4:** Wind direction is a natural global boundary descriptor for steady urban flow. It encodes the physical operating regime rather than providing unrealistic privileged information. In practice, this can be obtained from meteorological measurements or upstream sensors. For other PDE-based problems, this same mechanism would encode global physical parameters like inlet conditions or source terms.
>
> ---
> **Implementation Note:**
>
> Finally, we would also like to clarify one implementation detail raised in the weaknesses section. In the current benchmark, observations are placed on mesh nodes for controlled evaluation, but this is not a conceptual requirement of the framework. More generally, the method only requires an observation operator mapping measurements to the graph state; off-mesh sensors could be incorporated through interpolation/projection to nearby nodes or elements. We will make this distinction explicit in the revision, so that node-based placement is not interpreted as a fundamental scalability limitation.

---

> > ### Author Rebuttal · Reviewer_QgwV · 2026-04-02
> >
> > Thank authors for the detailed replies. Most of my concerns have been well addressed. But its lack of extensive comparison with modern DA methods remains as a central concern. It should be addressed before acceptance.

---

> > > ### Author Response · Authors · 2026-04-07
> > >
> > > We thank the reviewer for the helpful follow-up. To further strengthen the comparison with classical DA methods, we implemented an additional **reduced-order 3D-Var baseline** tailored to our static urban-flow benchmark.
> > >
> > > Our setting consists of **steady RANS snapshot reconstruction on geometry-varying irregular meshes**, rather than sequential forecast-assimilation in time. For this reason, standard full-state EnKF or 3D-Var formulations are not the most natural comparators, since no temporal forecast model is available and the state is defined on geometry-dependent unstructured meshes. We therefore implemented a classical DA baseline that is well matched to this benchmark: a **reduced-order 3D-Var method in a POD space** [1,2], with an explicit background prior and observation covariance.
> > >
> > > Specifically, we represent the state as
> > >
> > > $$
> > > x \\approx \\mu + \\Phi a
> > > $$
> > >
> > > and estimate the reduced coefficients by minimizing
> > >
> > > $$
> > > J(a)=\\tfrac{1}{2}(y-H(\\mu+\\Phi a))^\\top R^{-1}(y-H(\\mu+\\Phi a))+\\tfrac{1}{2}a^\\top B_a^{-1}a
> > > $$
> > >
> > > where $\\mu$ is the POD mean field, $\\Phi$ contains the retained POD modes, $a$ is the reduced coefficient vector, $R=\\sigma_{\\mathrm{obs}}^2 I$ is the observation covariance with $\\sigma_{\\mathrm{obs}}$ the observation-noise standard deviation, and $B_a$ is a diagonal background covariance estimated from the POD spectrum.
> > >
> > > The corresponding minimizer is
> > >
> > > $$
> > > a^\\star=\\left((H\\Phi)^\\top R^{-1}(H\\Phi)+B_a^{-1}\\right)^{-1}(H\\Phi)^\\top R^{-1}(y-H\\mu)
> > > $$
> > >
> > > $$
> > > x^\\star=\\mu+\\Phi a^\\star
> > > $$
> > >
> > > This is therefore a genuine **variational DA baseline with an explicit prior and observation model**. We tuned this baseline over several parameter configurations, including the reduced basis dimension and observation-noise/background-regularization settings, and report the best-performing configuration.
> > >
> > > Compared with LCSVD, this baseline makes the DA formulation explicit in the reduced coefficient space by combining an observation model with a background covariance, rather than estimating reduced coefficients via unregularized least-squares projection onto a truncated basis.
> > >
> > > The resulting comparison is:
> > >
> > > | Obs. | GenDA | 3D-Var |
> > > |---|---|---|
> > > | 300 | **0.312 / 0.823 / 0.937** | 0.375 / 0.814 / 0.909 |
> > > | 3000 | **0.192 / 0.867 / 0.976** | 0.368 / 0.819 / 0.913 |
> > > | 8000 | **0.137 / 0.898 / 0.987** | 0.367 / 0.820 / 0.914 |
> > >
> > > Each entry is reported as RRMSE / SSIM / cosine similarity.
> > >
> > > We will incorporate this new baseline in the revision and clarify why reduced-order 3D-Var is the most appropriate classical DA comparator for the present steady-snapshot benchmark. We hope this additional baseline helps address the remaining concern regarding comparison with classical DA methods.
> > >
> > >
> > >
> > > [1] Lorenc, A. C. (1986). Analysis methods for numerical weather prediction. Quarterly Journal of the Royal Meteorological Society, 112(474), 1177-1194.
> > >
> > > [2] Vermeulen, P. T. M., & Heemink, A. W. (2006). Model-reduced variational data assimilation. Monthly weather review, 134(10), 2888-2899.

---

### Official Review · Reviewer_NG7x · 2026-03-09

**Soundness:** 3
**Presentation:** 3
**Significance:** 3
**Originality:** 2
**Overall Recommendation:** 4
**Confidence:** 3

**Summary:**

This paper introduces GenDA, a generative data assimilation framework for reconstructing high-resolution urban wind fields from sparse sensor data on complex unstructured meshes. GenDA uses a multiscale graph diffusion model with classifier-free guidance, where an unconditional branch learns a geometry-aware flow prior and a sensor-conditioned branch injects observation constraints during sampling. On CFD data from a real Bristol neighborhood, it outperforms supervised GNN surrogates and a reduced-order DA baseline, especially at low sensor densities.

**Compliance With Llm Reviewing Policy:**

Affirmed.

**Final Justification:**

my concerns have been resolved, so I raise my raise to weak accept

**Key Questions For Authors:**

1.  Related to weakness 1, could you clarify how well GenDA transfers to truly different cities or layouts, and whether you have any preliminary results on such cross-city generalization?

2. A key selling point is that GenDA is generative and can produce ensembles for uncertainty-aware inference, but the evaluation focuses on pointwise metrics (RRMSE, SSIM, MAC) of reconstructions. Have you evaluated the calibration of the ensembles (CRPS), especially in sparse-sensor regimes where uncertainty is most critical?

**Limitations:**

Yes

**Strengths And Weaknesses:**

Strength:

1. Originality: The multiscale graph hierarchy (fine and coarsened meshes with o2o/o2r/r2r/r2o edges) is a sensible way to combine local obstacle-aware detail with efficient global information propagation over large, irregular domains.

2. Soundness: GenDA consistently improves RRMSE, SSIM, and directional similarity over supervised GNN surrogates  and over a classical reduced-order DA baseline.

3. Presentation: the paper is easy to follow and the method is straightforward. The visualization of the results are also very helpful.

Weakness:

1. Soundness: All experiments are on RANS simulations of one real neighborhood in Bristol, using 2D horizontal slices at a few altitudes; it is unclear to me that whether GenDA is able to generalization to other cities, flow regimes, or true 3D setups.

2. While the geometry-aware prior and CFD-based training do encode physics implicitly, there is no explicit enforcement of physical constraints.

3. The baselines are supervised GNNs and LCSVD; there is no direct comparison to other diffusion-based reconstruction methods, Therefore, it leaves open how much of the gain comes from a diffusion model versus the specific GenDA design.

---

> ### Author Rebuttal · Authors · 2026-03-31
>
> We thank the reviewer for the careful reading and for highlighting both the strengths of the multiscale graph design and the empirical gains of GenDA over the supervised baselines. We agree that the main questions are the scope of generalization beyond the current Bristol setting, the uncertainty-aware evaluation, and the need for a direct diffusion-based comparison. We address these points below.
>
> > Q1: Could you clarify how well GenDA transfers to truly different cities or layouts, and whether you have any preliminary results on such cross-city generalization?
>
> **A1:** We agree that the empirical scope of the current submission should be stated more explicitly. While we do not claim that cross-city or true 3D generalization is already established, we believe the present dataset is a stronger test of geometric generalization than a standard random split:
>
> * **Unseen 2D Geometries:** Our train/test split is performed across *altitude slices*, not across samples from a fixed 2D layout. Because obstacle footprints, street canyons, and flow pathways change with altitude, the model must generalize to new fluid-obstacle interaction patterns rather than simply interpolating a fixed map.
> * **Transfer to Other Cities:** The framework applies as long as the training data spans the target geometry/flow distribution. The graph construction, multiscale coarsening, and guided sampling are not tied to a specific city. We are currently generating additional CFD simulations for new urban layouts, but this is computationally costly and time-consuming.
> * **Extension to 3D:** The core method is not inherently restricted to 2D slices; the graph-based prior learning and guided reconstruction naturally extend to volumetric meshes. The main challenges are computational scale and training-data cost, rather than conceptual limitations.
>
> We will revise the paper to make this scope explicit. We will also clarify that the present model learns physics implicitly (via the CFD training distribution) and does not yet include explicit PDE-residual or divergence constraints; incorporating these is an important future direction.
>
> > Q2: A key selling point is that GenDA is generative and can produce ensembles for uncertainty-aware inference, but the evaluation focuses on pointwise metrics. Have you evaluated the calibration of the ensembles (CRPS), especially in sparse-sensor regimes where uncertainty is most critical?
>
> **A2:** Following the reviewer’s suggestion, we evaluated held-out sensor CRPS under the same masked-sensor protocol used for the new interpolation experiment requested by reviewer Z6xY. This gives a probabilistic score directly at unseen sensor locations. The result is a clear monotonic trend: as the number of observations before masking increases, the held-out sensor CRPS decreases, indicating that the predictive ensembles become progressively better calibrated and sharper as more observational information is available:
>
> * **Held-out CRPS Trend:** ~0.058 $\rightarrow$ 0.055 $\rightarrow$ 0.048 $\rightarrow$ 0.040 (as observations increase).
>
> Beyond the CRPS metrics, we also evaluated the field-level distributional structure. We provide an aggregate density comparison for the 300-observation regime here: https://anonymous.4open.science/r/ICML_8122_rebuttal_NG7x-1D23/density_aggregate_obs300.png.
>
> This figure plots the predicted and ground-truth marginal PDFs of $U_x$ and $U_y$ aggregated across test cases. In this highly sparse regime, GenDA's predicted ensemble distribution remains closely aligned with the target marginals, as reflected by the Wasserstein distance (where lower indicates closer agreement):
>
> * **Wasserstein Distance (300 obs):** 0.0150 for $U_x$ | 0.0120 for $U_y$.
>
> This complements the pointwise CRPS evaluation, confirming that the model captures realistic field-level uncertainty and distributional structure even with very limited observations. We will explicitly highlight these uncertainty-aware capabilities in the revision.
>
> ---
> **Note on Diffusion Baselines:** Regarding the concern about direct diffusion-based comparisons, please refer to our response to **Reviewer 9Lnk**, where we add a new stochastic diffusion baseline and discuss that comparison in detail.

---

> > ### Author Rebuttal · Reviewer_NG7x · 2026-04-04
> >
> > my concerns have been resolved, so I raise my raise to weak accept

---

> > > ### Author Response · Authors · 2026-04-07
> > >
> > > We thank the reviewer for the thoughtful feedback and positive reassessment. We are glad that the added clarifications helped address the concerns. We will incorporate these points clearly in the revision.

---

### Official Review · Reviewer_9Lnk · 2026-03-12

**Soundness:** 3
**Presentation:** 3
**Significance:** 3
**Originality:** 2
**Overall Recommendation:** 4
**Confidence:** 4

**Summary:**

This paper studies the problem of reconstructing wind fields from sparse sensor observations in complex urban geometries. The authors propose GenDA, a diffusion-based generative framework that models a geometry-aware prior of urban wind fields on unstructured meshes and incorporates sensor observations during sampling via classifier-free guidance. Experiments are conducted on simulated data from the Bristol urban area and compared with graph neural network methods (MGN, MS-MGN) and a reduced-order data assimilation method (LCSVD). The results show that GenDA achieves better reconstruction performance across multiple evaluation metrics, with particularly noticeable improvements under sparse observation settings.

**Compliance With Llm Reviewing Policy:**

Affirmed.

**Final Justification:**

Overall, I find the problem setting, performing data assimilation on high-resolution wind fields using an unstructured grid, both novel and meaningful, and I believe the proposed method is well motivated. Some experimental findings, such as the performance degradation with an increasing number of observations, could have been investigated in greater depth. However, this limitation does not undermine the paper’s overall contribution. Therefore, I would recommend a weak accept.

**Key Questions For Authors:**

In Table 1, the SSIM of MGN slightly decreases as the number of observations increases. This seems counterintuitive, since higher observation density would typically improve reconstruction quality. Could the authors clarify why this occurs (e.g., variance across random sensor placements, metric sensitivity, or training/evaluation mismatch)?

How much does the proposed multiscale graph representation contribute to the final assimilation/reconstruction quality? For example, if one keeps the CFG framework unchanged but replaces the multiscale graph design with a simpler (single-scale) network, how do the results change?

All three baselines appear to be deterministic. How does GenDA compare to other stochastic/generative data assimilation approaches (e.g., score-based data assimilation method [1] or DiffDA [2])? Including at least one additional stochastic baseline will make the contribution more solid.

[1] Rozet, François, and Gilles Louppe. "Score-based data assimilation." Advances in Neural Information Processing Systems 36 (2023): 40521-40541.

[2] Huang, Langwen, et al. "Diffda: a diffusion model for weather-scale data assimilation." arXiv preprint arXiv:2401.05932 (2024).

**Limitations:**

Yes.

**Strengths And Weaknesses:**

**Strengths:**

The method builds on a multiscale graph neural architecture over unstructured meshes. This design is well-suited to complex urban geometries and appears to help preserve salient flow structures such as wakes and recirculation zones.

The experimental setup includes multiple observation regimes (different sensor configurations), which helps demonstrate robustness to changes in measurement layout and sparsity.

The evaluation is fairly comprehensive, using several complementary metrics to assess reconstruction quality.

**Weaknesses:**

The choice of the guidance weight $\gamma$ is not supported by a clear, principled selection strategy. Since $\gamma$ controls the trade-off between the learned prior and observation consistency, it would be helpful to provide either practical guidelines (e.g., a rule-of-thumb), sensitivity analysis, or a discussion linking $\gamma$ to observation noise/reliability.

The baseline suite is somewhat limited, and a few results are difficult to interpret. This may lead readers to question whether the baselines are implemented and tuned competitively (see questions below). Stronger or more carefully validated baselines would make the empirical claims more convincing.

The novelty may be incremental relative to prior diffusion guidance techniques. Classifier-free guidance (CFG) is well-established; the primary contribution appears to be adapting it to unstructured-mesh flow reconstruction via the proposed multiscale mesh/graph representation. A more systematic ablation would strengthen the paper by isolating the contribution of each component (e.g., multiscale graph representation, conditioning design, and guidance mechanism) and clarifying which elements are essential for the reported gains.

---

> ### Author Rebuttal · Authors · 2026-03-31
>
> We thank the reviewer. To address the concerns about guidance-weight selection, baseline strength, and component-wise contribution, we performed three additional analyses: (i) an expanded guidance-weight sweep, (ii) a single-mesh GenDA ablation, and (iii) a new stochastic diffusion baseline.
>
> **Guidance-weight selection:** Appendix C.1 already shows that intermediate guidance values work best, while overly large guidance can hurt reconstruction. The expanded sweep makes the rule explicit. At 300 observations, RRMSE improves from 0.311 at $\gamma=1.0$ to 0.303 at $\gamma=1.5$ and 0.298 at $\gamma=2.0$, with only marginal gain at $\gamma=3.0$ (0.296). This does not persist at higher densities: at 3000 observations, RRMSE is 0.193 / 0.184 / 0.183 / 0.291 for $\gamma=1.0/1.5/2.0/3.0$; at 8000, it is 0.137 / 0.131 / 0.145 / 0.481. SSIM and cosine similarity show the same pattern: $\gamma=2$ stays in the best regime across densities, while $\gamma=3$ becomes unstable as observational information increases. Thus, $\gamma=2$ was chosen not for isolated peak performance, but for robustness across densities, slices, and wind directions.
>
> > Q1: In Table 1, the SSIM of MGN slightly decreases as the number of observations increases. This seems counterintuitive. Could the authors clarify why this occurs?
>
> **A1:** We do not believe this is caused by the evaluation protocol, since all baselines use the same sampled sensor sets and seeds. Our new single-mesh GenDA ablation clarifies this directly. This model uses the same single-scale mesh as MGN, but within the generative DA framework. In that case, SSIM increases monotonically with observation density: $0.791 \rightarrow 0.830 \rightarrow 0.897$ for 300 / 3000 / 8000 observations. In contrast, MGN slightly decreases: $0.678 \rightarrow 0.642 \rightarrow 0.633$. This suggests the issue is not the sensor protocol, but the deterministic supervised reconstruction used by MGN on the single mesh. We believe MGN tends to produce smoother, mean-biased reconstructions, and SSIM is sensitive to such structural over-smoothing. MS-MGN is also informative: it improves to $0.655 / 0.669 / 0.681$, showing that better long-range propagation helps, but much less than single-mesh GenDA and full GenDA.
>
> > Q2: How much does the proposed multiscale graph representation contribute? How do results change with a single-scale network?
>
> **A2:** To isolate this, our **single-mesh GenDA ablation** retains the exact classifier-free guidance (CFG) framework but removes the reduced mesh. We observe clear degradation. At 300 observations, RRMSE worsens from 0.312 (GenDA) to 0.399 (single-mesh GenDA), and Cosine similarity drops from 0.937 to 0.899. At 3000 observations, RRMSE degrades from 0.192 to 0.254, and Cosine from 0.976 to 0.959.
>
> This confirms the multiscale graph's dual role. Computationally, the reduced mesh enables deep, long-range message passing; representationally, the original mesh preserves local, obstacle-aware geometry. The hierarchy is vital for coarse-scale communication and fine-scale correction, which we will explicitly highlight.
>
> > Q3: How does GenDA compare to other stochastic/generative DA approaches (e.g., score-based DA [1] or DiffDA [2])?
>
> **A3:** We agree that a stronger stochastic comparison is valuable. However, the cited methods are not plug-and-play baselines for our method. Score-based DA targets time-evolving Markovian systems via posterior sampling, whereas we reconstruct static fields on irregular meshes. DiffDA assumes structured atmospheric grids and strong background forecasts, unlike our sparse node observations on geometry-varying unstructured meshes.
>
> However, to provide a strong stochastic baseline as recommended by the reviewer, we trained **MultiMesh-VP**: it uses GenDA's exact multiscale backbone and standard VP diffusion [1] conditioned directly on observations, bypassing our unconditional/conditional formulation and CFG posterior correction.
>
> **Performance Comparison (RRMSE / SSIM / Cosine):**
>
> | Obs | GenDA | MultiMesh-VP | MGN | MS-MGN | LCSVD |
> | :--- | :--- | :--- | :--- | :--- | :--- |
> | **300** | **0.312 / 0.823 / 0.937** | 0.415 / 0.706 / 0.907 | 0.426 / 0.678 / 0.885 | 0.412 / 0.655 / 0.899 | 0.360 / 0.831 / 0.915 |
> | **3000** | **0.192 / 0.867 / 0.976** | 0.258 / 0.777 / 0.962 | 0.382 / 0.642 / 0.905 | 0.333 / 0.669 / 0.935 | 0.354 / 0.834 / 0.918 |
> | **8000** | **0.137 / 0.898 / 0.987** | 0.189 / 0.822 / 0.978 | 0.318 / 0.633 / 0.932 | 0.277 / 0.681 / 0.954 | 0.354 / 0.834 / 0.918 |
>
> MultiMesh-VP easily beats the deterministic GNNs but strictly underperforms GenDA. This isolates our core contribution: the gains stem not just from multiscale message passing or baseline diffusion, but specifically from our formulation of learning a geometry-aware prior and using CFG for observation-consistent posterior reconstruction.
>
> *[1] Song, Yang, et al. "Score-based generative modeling through stochastic differential equations." arXiv (2020).*

---

> > ### Author Rebuttal · Reviewer_9Lnk · 2026-04-03
> >
> > Thanks for the detailed response. I will maintain my original positive score.

---

> > > ### Author Response · Authors · 2026-04-07
> > >
> > > We thank the reviewer for the helpful suggestions, which helped improve the quality of the paper. We will incorporate the additional analyses and clarifications in the revision.

---

### Official Review · Reviewer_Z6xY · 2026-03-12

**Soundness:** 3
**Presentation:** 3
**Significance:** 3
**Originality:** 2
**Overall Recommendation:** 4
**Confidence:** 3

**Summary:**

This paper addresses the problem of reconstructing high-resolution urban wind fields from sparse sensor measurements over complex geometries. The authors formulate the task as a conditional generative modeling problem and propose GenDA, a multiscale graph-based diffusion framework for data assimilation on unstructured meshes.

The model is trained on Reynolds-averaged Navier–Stokes simulations of a real urban neighborhood. An unconditional branch learns a geometry-conditioned prior over flow fields, while a sensor-conditioned branch incorporates sparse observations. During inference, classifier-free guidance combines the two branches; the authors interpret this mechanism as approximate tempered posterior sampling.

The architecture operates on large unstructured meshes using a two-level multiscale graph structure that enables both local message passing on the original mesh and long-range propagation on a coarsened mesh. The framework supports multiple sensing configurations, including random sampling, clustered sensor regions, and trajectory-based observations.

Experiments are conducted on held-out altitude slices (unseen geometries) and across varying sensor densities. The method is compared against supervised graph neural network baselines and a classical reduced-order data assimilation approach. Quantitative results reported in terms of relative RMSE, structural similarity , and cosine similarity indicate improved reconstruction accuracy, particularly in sparse-observation regimes. Additional analyses examine the effect of guidance strength and observation density.

**Compliance With Llm Reviewing Policy:**

Affirmed.

**Final Justification:**

After reading the rebuttal and follow-up discussion, I appreciate the authors’ detailed and constructive responses.

In particular, the addition of the masked-sensor validation experiment meaningfully strengthens the empirical evaluation. The reported results provide direct evidence of prediction accuracy at unseen sensor locations and address an important aspect of the model’s practical behavior. This improves my confidence in the experimental rigor of the work.

The authors also provide a clearer and more precise discussion of the intended scope of generalization. I appreciate the explicit acknowledgment that the current submission demonstrates the framework in a single application domain and that broader applicability remains a direction for future work.

Overall, the rebuttal improves the clarity and strengthens parts of the evaluation. However, my main concern regarding the demonstrated breadth of applicability remains, as the empirical validation is still limited to a single domain and cross-domain generalization is not established. In addition, the methodological contribution continues to lie primarily in the integration of existing components rather than in new algorithmic developments.

Taking these points into account, my overall assessment remains unchanged.

**Key Questions For Authors:**

1. Scope of Generalization

The empirical evaluation focuses on urban wind reconstruction derived from RANS simulations of a single urban environment. While the paper suggests that the framework could apply more broadly to inverse problems on unstructured meshes, this broader applicability is not empirically demonstrated.

Could the authors clarify the intended scope of generalization of the proposed approach? In particular:
a), To what extent does the method depend on properties specific to steady 2D RANS flow fields?
b), Would the framework be directly applicable to other data assimilation settings, such as different PDE-governed systems (e.g., weather prediction, subsurface flow, structural mechanics), or to time-dependent problems?
c) What assumptions about geometry, mesh structure, or data distribution are critical for the method to work effectively?

A clearer discussion of generalization conditions and potential target domains would help assess the broader impact and significance of the contribution. Depending on the response, this could positively influence my evaluation of the paper’s significance beyond the specific application studied.

2. Masked-Sensor Validation Experiment

The current evaluation primarily measures full-field reconstruction error over the entire mesh. I did not observe an experiment in which, under a fixed sensor configuration, a subset of sensors is intentionally masked at test time and the model’s predictions are evaluated specifically at those held-out sensor locations.

Have the authors considered performing such a masked-sensor validation experiment? For example:

Step 1. Fix a sensor network configuration,
Step 2. Withhold a subset of sensor measurements during inference,
Step 3. Evaluate prediction accuracy at those withheld locations using pointwise error or statistical metrics.

This type of evaluation would provide a direct measure of interpolation capability and predictive accuracy at unseen measurement points, which is particularly relevant for practical data assimilation scenarios. If such results are available (or could be provided), they would strengthen the empirical validation and could influence my assessment of the experimental rigor and practical relevance of the method.

**Limitations:**

The authors discuss several technical boundaries of their work, including the focus on steady 2D urban wind fields and possible extensions to temporal or volumetric settings. However, the limitations could be articulated more explicitly in two areas.

1. While the method is presented as a general framework for generative data assimilation on unstructured meshes, empirical validation is confined to a single urban wind setting. A clearer discussion of the assumptions required for generalization to other PDE-based systems or data assimilation tasks would strengthen the reflection on scope.

2. The methodological contribution primarily lies in the integration of established components (EDM diffusion, multiscale MeshGraphNet architectures, and classifier-free guidance) rather than in new architectural or algorithmic primitives. While this does not affect technical soundness, a more explicit acknowledgment of this positioning would clarify the nature of the contribution.

**Strengths And Weaknesses:**

Soundness
The submission is technically sound. The formulation of data assimilation as conditional generative modeling with diffusion and classifier-free guidance is coherent, and the interpretation of guidance as approximate tempered posterior sampling is clearly motivated. The architectural choices (EDM-style diffusion and multiscale MeshGraphNet-style message passing) are appropriate for large unstructured meshes.

The experimental evaluation is generally well designed. The method is tested on held-out geometries, varying sensor densities, and multiple sensing strategies, and compared against both supervised GNN baselines and a classical reduced-order data assimilation method. Results are reported using complementary metrics (RRMSE, SSIM, MAC), and implementation details are sufficiently documented in the appendix.

One limitation is that evaluation focuses mainly on full-field reconstruction error. An additional experiment that masks part of a fixed sensor network at test time and evaluates prediction accuracy at those held-out locations could provide a more direct assessment of interpolation performance.

Presentation
The paper is clearly written and logically structured. The methodological components are described with adequate detail, and the appendices provide sufficient information for reproducibility. The work is positioned within related literature on diffusion models, graph-based simulation, and data assimilation.

A clearer discussion of the broader applicability of the approach beyond the specific urban wind setting would strengthen the framing.

Significance

The paper addresses data assimilation in high-dimensional PDE-based systems on unstructured meshes, which is relevant to scientific machine learning. The empirical results show consistent improvements over baselines, particularly in sparse-observation regimes.

However, the demonstrated scope is limited to a single application domain (urban wind reconstruction). While broader applicability is suggested, it is not empirically validated, which may limit the perceived breadth of impact.

Originality

The core components of the method (EDM diffusion, multiscale graph networks, classifier-free guidance) are established techniques. The contribution primarily lies in their integration for data assimilation on large unstructured meshes and in the interpretation of guidance as posterior tempering in this context.

The originality therefore stems from the formulation and system-level combination of existing methods rather than from new algorithmic developments.

---

> ### Author Rebuttal · Authors · 2026-03-31
>
> We thank the reviewer for the assessment. We appreciate the suggestion to include a masked-sensor evaluation, a direct interpolation test.
>
> > Q1: Could the authors clarify the intended scope of generalization of the proposed approach, including dependence on steady 2D RANS fields, applicability to other PDE/data-assimilation settings and time-dependent problems, and critical assumptions on geometry, mesh structure, or data distribution?
>
> **A1:** We agree the scope should be stated more explicitly. Our aim is twofold: (1) address urban wind-field reconstruction, relevant to air quality, pollutant dispersion, pedestrian comfort, and risk-aware planning; and (2) use this setting as a concrete testbed for a more general framework for generative DA on unstructured meshes.
>
> The empirical validation is limited to steady urban wind reconstruction from 2D slices of 3D RANS simulations of a neighborhood. However, the core formulation is not specific to RANS. Its general components are: (i) representing the state on an unstructured mesh/graph, (ii) learning a geometry-aware prior through the unconditional branch, and (iii) incorporating sparse observations at inference through classifier-free guidance. Application-specific elements are the training distribution and conditioning variables.
>
> In our setting, global wind direction is included in the conditioning vector together with the diffusion noise level. We view it as a global descriptor of the physical instance, analogous to a boundary condition, forcing, or operating regime. For other PDE-governed systems, the same mechanism could encode inlet/outlet conditions, source terms, forcing amplitudes, material coefficients, loading conditions, or other global descriptors. Thus, the conditioning mechanism is not specific to urban wind reconstruction; it is a general way to inject problem-defining information into the prior and guided reconstruction.
>
> More broadly, we expect transfer when: (1) the state can be represented on a mesh/graph, (2) observations can be mapped to that representation, and (3) the training data sufficiently covers the target distribution to learn a meaningful prior. The paper demonstrates this in one domain rather than claiming broad empirical validation across PDE families.
>
> For time-dependent problems, the main change would be temporal modeling rather than the graph-based assimilation mechanism itself. A natural extension is to condition the denoiser on previous assimilated states and current observations autoregressively, or to model short temporal windows jointly. In both cases, the geometry-aware graph representation and observation-guidance mechanism remain central. We agree that temporal and cross-domain generalization are not empirically established in the current submission and will clarify this in the revision.
>
> > Q2: Have the authors considered a masked-sensor validation experiment, e.g., fixing a sensor network, withholding a subset of measurements at inference, and evaluating accuracy at those withheld locations?
>
> **A2:** Following the reviewer’s suggestion, we performed a new held-out-sensor experiment. For each test case, we sample a fixed sensor configuration of size N, retain only 25% of those measurements during inference, and evaluate on the remaining 75% held-out sensor nodes. This measures prediction quality at unseen sensor locations under a fixed network.
>
> Held-out sensor RRMSE (mean ± std):
>
> |        | GenDA | MGN | MS-MGN | LCSVD |
> |--------|------:|----:|-------:|------:|
> | 300 obs  | 0.318 ± 0.101 | 0.444 ± 0.155 | 0.436 ± 0.151 | 0.346 ± 0.131 |
> | 3000 obs | 0.217 ± 0.068 | 0.413 ± 0.146 | 0.363 ± 0.121 | 0.334 ± 0.125 |
> | 8000 obs | 0.160 ± 0.049 | 0.359 ± 0.131 | 0.309 ± 0.099 | 0.333 ± 0.121 |
>
> Trend plots and qualitative examples for 300 and 3000 observations are provided at https://anonymous.4open.science/r/ICML_8122_rebuttal_Z6xY-9350/figures.md. These results directly address the suggested masked-sensor validation and confirm that GenDA improves not only full-field reconstruction, but also prediction quality at held-out sensor locations.
>
> **Limitations:** We agree the limitations should be stated more explicitly. First, the empirical validation is restricted to one application domain—steady urban wind reconstruction from RANS-derived slices—and does not by itself establish broad cross-domain generalization. We will revise the paper to make this scope boundary clearer and state the assumptions for transfer to other PDE-based DA settings. Second, our contribution is not a new diffusion objective or graph-learning primitive; rather, it is a framework formulation integrating diffusion modeling, multiscale graph message passing, and classifier-free guidance for DA on unstructured meshes. We believe this is meaningful: it enables generative DA on large irregular domains and yields consistent improvements over supervised graph baselines and a classical reduced-order DA baseline, especially in sparse-observation regimes.

---

> > ### Author Rebuttal · Reviewer_Z6xY · 2026-04-06
> >
> > The authors provide a clear and constructive rebuttal and address several of my concerns.
> >
> > Regarding generalization (Q1), the authors clarify the intended scope of the framework and provide a more detailed discussion of how the approach could extend to other PDE-governed systems. This explanation is helpful and improves my understanding of the method’s broader applicability. However, the validation remains limited to a single application domain, and the broader generalization claims are still primarily conceptual rather than empirically demonstrated. I view this concern as partially resolved.
> >
> > Regarding the masked-sensor validation (Q2), I appreciate that the authors conducted an additional experiment following my suggestion. The reported results provide a more direct evaluation of prediction accuracy at unseen sensor locations and strengthen the empirical validation of the method. This concern is largely resolved.
> >
> > Overall, the rebuttal improves the clarity of the paper and strengthens parts of the evaluation. My remaining concerns are mainly about the demonstrated breadth of applicability rather than technical soundness.

---

> > > ### Author Response · Authors · 2026-04-07
> > >
> > > We thank the reviewer for the thoughtful follow-up. We agree that the current submission demonstrates the framework in a single application domain, and that broader cross-domain generalization is not empirically established by the present experiments. We will revise the paper to make this scope boundary explicit and to frame broader applicability as a motivated extension rather than a demonstrated claim.
> > >
> > > At the same time, we hope the contribution remains meaningful under this narrower framing. Our intended claim is not that we have already validated a universally general DA method across PDE families, but that we provide a framework-level demonstration that generative data assimilation can be made effective on large irregular domains represented by unstructured meshes, enabling the reconstruction of complex urban flow structures such as wakes and recirculation patterns from sparse observations. We view the urban wind setting as a practically relevant and technically challenging testbed for this question, rather than as the sole intended scope of impact.
> > >
> > > In the revision, we will therefore (1) narrow the wording around generalization, (2) state explicitly the assumptions under which transfer to other PDE-based DA settings may be expected, and (3) position broader applicability as an important future direction rather than an empirically verified conclusion. We appreciate that your feedback helped us better calibrate the framing of the paper.

---

### Decision · Program_Chairs · 2026-04-30

**Decision:**

Accept (regular)

**Comment:**

Three of the four reviewers supported acceptance, while one maintained a reject recommendation. After the rebuttal, two reviewers said that their concerns had been resolved, and the other kept the same weak accept assessment. Across the reviews, the main concerns were limited methodological novelty, evaluation being limited to a single urban wind setting, and the need for stronger comparisons or clearer validation of some design choices. At the same time, reviewers generally viewed the paper as technically sound overall, clearly written, and showing consistent empirical gains over the reported baselines, especially under sparse observations. On balance, I view the paper as acceptable, mainly because the method is well motivated and the empirical case is sufficient for acceptance, even though the broader scope and novelty should be stated more carefully.